# Miniature spatial transcriptomics for studying parasite-endosymbiont relationships at the micro scale

Hailey Sounart [1,3], Denis Voronin[2,3], Yuvarani Masarapu [1], Matthew Chung[2], Sami Saarenpää [1], Elodie Ghedin [2,4] ✉ & Stefania Giacomello [1,4] ✉

Several important human infectious diseases are caused by microscale-sized parasitic nematodes like filarial worms. Filarial worms have their own spatial tissue organization; to uncover this tissue structure, we need methods that can spatially resolve these miniature specimens. Most filarial worms evolved a mutualistic association with endosymbiotic bacteria *Wolbachia*. However, the mechanisms underlying the dependency of filarial worms on the fitness of these bacteria remain unknown. As *Wolbachia* is essential for the development, reproduction, and survival of filarial worms, we spatially explored how *Wolbachia* interacts with the worm's reproductive system by performing a spatial characterization using Spatial Transcriptomics (ST) across a posterior region containing reproductive tissue and developing embryos of adult female *Brugia malayi* worms. We provide a proof-of-concept for miniature-ST to explore spatial gene expression patterns in small sample types, demonstrating the method's ability to uncover nuanced tissue region expression patterns, observe the spatial localization of key *B. malayi* - *Wolbachia* pathway genes, and co-localize the *B. malayi* spatial transcriptome in *Wolbachia* tissue regions, also under antibiotic treatment. We envision our approach will open up new avenues for the study of infectious diseases caused by micro-scale parasitic worms.

Lymphatic filariasis (elephantiasis) and onchocerciasis (river blindness) are two debilitating and far-reaching neglected tropical infectious diseases affecting more than 150 million of "the poorest of the poor" worldwide[1–3]. The parasitic nematodes that cause these diseases —*Wuchereria bancrofti, Brugia malayi*, and *B. timori* for lymphatic filariasis, and *Onchocerca volvulus* for onchocerciasis—result in, respectively, lymphoedema that can progress to elephantiasis, or blindness[3]. Mass drug administration campaigns of affordable and safe microfilaricidal drugs are used to halt parasite transmission by killing the microfilariae (mf) in the blood or the skin of the infected hosts.

However, the long lifespan of the adult worms (8 years for lymphatic filariasis and 20-30 years for river blindness) requires repeated microfilaricidal treatment. Therefore, alternative therapeutic approaches that affect the adult worms are critically needed.

*Wolbachia* are common intracellular bacteria found in arthropods and filarial nematodes[4]. Filarial parasites that cause lymphatic filariasis and onchocerciasis have evolved a mutualistic association with *Wolbachia* that is essential for worm development, reproduction, and survival[5]. The endosymbiont can be eliminated from the worms by treating infected animals and humans with a small variety of antibiotics

[1]Department of Gene Technology, KTH Royal Institute of Technology, SciLifeLab, Stockholm, Sweden. [2]Systems Genomics Section, Laboratory of Parasitic Diseases, National Institute of Allergy and Infectious Diseases, National Institutes of Health, Bethesda, MD, USA. [3]These authors contributed equally: Hailey Sounart, Denis Voronin. [4]These authors jointly supervised this work: Elodie Ghedin, Stefania Giacomello. ✉e-mail: elodie.ghedin@nih.gov; stefania.giacomello@scilifelab.se

(such as doxycycline and rifampicin), which in turn results in the death of the adult worms, ultimately curing the mammalian host of the filarial infection[4,6–9]. Doxycycline treatment demonstrates strong effects on the embryogenesis of filarial nematodes, causing apoptosis in developing embryos after 6 days of treatment in vitro[10], and eventual embryo clearance from the uterus of females treated in vivo during clinical trials[3,11,12]. *B. malayi* became the model organism for studying the molecular processes of disease-causing filarial worms as it can develop from later larval (L3) to adult life cycle stages in the laboratory animal model *Meriones unguiculatus*[13]. A number of studies explored at the molecular level the co-dependencies between *B. malayi* and its *Wolbachia* (*w*Bm)[14–21]. Homeostasis of the mutualistic relationship that evolved between *B. malayi* and *w*Bm requires the coordinated regulation of *B. malayi* genes. This dependency on *Wolbachia* for oogenesis and embryogenesis in female worms means that therapeutic approaches that deplete *w*Bm in the *B. malayi* worms can cause permanent sterilization of adult females. Targeting specific processes in *B. malayi* that are dependent on *Wolbachia* is thus a promising therapeutic direction because of the large impact *w*Bm has on the processes of oogenesis and embryogenesis within the reproductive tissue of adult female *B. malayi* worms, and evidence that antibiotic-induced elimination of *Wolbachia* results in the eventual death of the adult worms. However, much remains unknown about these interactions. Previous studies addressing interactions between *B. malayi* and *w*Bm[14–21] have used various targeted methods for specific sets of transcripts or proteins, or performed bulk RNAseq. A more detailed understanding of the *B. malayi*-*Wolbachia* relationship could come from approaches that resolve gene expression of these interactions at the level of individual tissue morphological regions in an unbiased, exploratory manner. Shedding light on the molecular mechanisms occurring in specific regions of the worm impacted by *Wolbachia* (such as reproductive tissues) and how such processes are distributed within the different tissue structures could provide insights into the biology of *B. malayi*, and the symbiosis with *Wolbachia*. This in turn could aid in the long-term goal of pinpointing more potent drug targets for lymphatic filariasis.

In recent studies of other infectious diseases caused by small pathogens, such as *Mycobacterium tuberculosis*[22], *Trypanosoma brucei*[23], and *Plasmodium berghei*[24], a variety of spatial and/or single cell approaches have been used to localize the organisms within host tissues and to study the host response to infection. However, some pathogens, such as multicellular parasitic worms, have their own tissue structure organization that has yet to be spatially resolved in an unbiased fashion. In the context of studies focusing on whole *B. malayi* adult parasites, recent transcriptomic and proteomic analyses have led to significant insight into the biology of the worm[25,26]. While comparative proteomic analysis of differences between the reproductive tissue, body wall, and digestive tract regions provided essential information for the whole tissue[25], nuanced expression patterns in specific regions of each tissue were likely missed. An exploratory approach that investigates transcriptome-wide expression patterns in whole tissue sections without the need for labor intensive dissection of different tissues would complement such proteomic efforts. In addition, there are specific regions of the worms that are systematically underrepresented in whole-parasite omics, such as the head region of the parasites that contains critical tissues at the host-parasite interface, and posterior regions where oogenesis and early embryogenesis occur and where *Wolbachia* are also located. Prioritizing omic analyses of these regions will improve our understanding of worm biology, life cycle, transmission, and the interface of parasite-host interactions. Thus far, only one study showed spatial-like transcriptomic analysis of the *B. malayi* head region using a combination of methods such as low-input tissue capture and RNA tomography with light-sheet and electron microscopy[26]. Similar efforts on *C. elegans*[27], *Pristionchus pacificus*[28], *Drosophila* embryos[29], and mouse embryos[30] also collected serial, transverse sections along the anterior-posterior (A/P) axis of the organism and performed RNAseq on each section[26,28–30]. In one study, such an approach was combined with laser capture microdissection (LCM) to isolate specific regions[30]. While these methods resolve transcripts along the A/P axis, the different tissue structures within each individual tissue section are not resolved, unless more labor intensive techniques such as LCM[26,30] or dissection[25] are employed. Alternative approaches leveraged isolating single cells, such as the first cell atlas of the parasitic worm *Schistosoma mansoni* using single-cell technology[31], or resolving mouse embryos by performing single-cell RNAseq within spatial spots[32]. However, isolating single cells for filarial nematodes such as *B. malayi* is complicated by the large hypodermal syncytial cells spanning the entire length of the worm body, making this strategy unfeasible. Single nucleus RNA sequencing offers the possibility of overcoming the anatomical constraints of a single-cell based strategy, however, crucial information provided by having the spatial context is lost with such a method. A recent study produced a high resolution, exploratory spatial map of *Drosophila* embryos[33], highlighting the importance of exploratory spatial approaches in uncovering tissue region specific patterns.

Spatial Transcriptomics (ST) is a high-throughput, sequencing-based exploratory method where polyadenylated transcripts are captured by spatially barcoded probes on a slide underneath a tissue section[34,35]. ST connects tissue morphology to gene expression by overlaying the spatially resolved transcriptome at 55 µm resolution on hematoxylin and eosin stained images[34,35]. This technique has been used to study a variety of tissue types and disease states primarily across human and mouse tissues[36], as well as some plant tissues[37,38], typically in the 1-6 mm range.

To enable spatially resolved transcriptomic methods for parasitic worms at the micro-scale, we focused on *B. malayi* as our model organism. We profiled adult female worms that are on average 150 µm in diameter and 43-55 mm in length and focused specifically on the often-understudied posterior region, important in the *B. malayi*-*w*Bm relationship, containing ovary tissue, the beginning of the uterus with fertilized eggs and early embryos, the digestive tract, and the body wall. We worked with intact, longitudinal tissue sections and resolved distinct structures spatially in an exploratory, whole transcriptome manner. Each longitudinal tissue section showed a 2D spatial layout of transcripts; the combined gene expression information from all 2D sections could be merged into a 3D model that reconstructed the region of interest where transcripts can be explored along multiple axes (anterior-posterior, ventral-dorsal). We have overcome the technical challenges of working with very small tissue sizes (micrometer-scaled) with particularly thin diameters (<200 µm) by developing sample preparation (fixation and embedding), cryosectioning, and tissue attachment/staining techniques for miniature Spatial Transcriptomics (miniature-ST). We used the miniature-ST method to uncover tissue-specific gene expression patterns in *B. malayi*, localize the expression of key glycolytic pathway genes, and co-localize the expression of *B. malayi* genes in tissue areas with and without *Wolbachia*. Finally, we used the approach on doxycycline-treated worms to evaluate the expression of *B. malayi* genes in response to anti-*Wolbachia* treatment. By developing this highly reproducible miniature-ST method, we are forging a research path to perform spatial characterization of gene expression profiles across tissues of small parasitic worms. A tissue-resolution level understanding of parasitic worm biology could help the development of more targeted therapeutic strategies.

## Results

### Miniature-ST provides reproducible capture of gene expression information

To visualize the spatial localization of *B. malayi* genes in the adult female worm, we first determined if spatial transcriptomics could be

applied to intact, longitudinal tissue sections from an organism of this size (~130-170 µm in diameter). We adapted the ST technology Visium Spatial Gene Expression assay (10X Genomics)[39] for unbiased capture of gene expression information. To apply ST to this small sample type, we faced several technical challenges and developed strategies to overcome these issues in terms of sample preparation, cryosectioning, and tissue attachment to the slides. For sample preparation, we developed several steps to improve downstream cryosectioning and tissue attachment to the Visium slides. These steps involved: i) methanol fixation before embedding the tissue in the Optimal Cutting Temperature (OCT) compound to improve tissue attachment, ii) hematoxylin staining before OCT embedding to visualize the clear worms in the embedded block, and iii) dissecting out smaller, i.e. ~5 mm, pieces to obtain intact cryosections during sectioning. When cryosectioning the worm tissue, it was difficult to obtain intact, longitudinal sections from such a small specimen (less than 20 total sections per sample). We observed that the sample needed to be completely flat to get usable sections through the embedded worm piece. Re-embedding of the small ~5 mm worm specimen onto flattened OCT facilitated acquiring multiple, intact sections throughout a single worm specimen. All sections from a sample could be placed on a single Visium slide capture area, reducing experimental costs (Fig. 1A). To determine if our re-embedding approach could affect the gene expression information captured, we included a sample, BM3, that was not re-embedded prior to sectioning. We found that the difference in the embedding technique did not impact the quality of the gene expression information captured as the gene expression patterns were reproducible between samples, but rather the re-embedding technique greatly aided in obtaining intact tissue sections (Fig. 1B-D). In addition, the RNA integrity number (RIN), a measure of the RNA quality, was similar for the originally embedded worms (6-8 RIN) and re-embedded worms (7-8 RIN), indicating the re-embedding did not impact RNA quality (Supplementary Table 1). However, the re-embedding did improve the orientation of the sample block for obtaining higher quality sections. A section thickness of 8 µm allowed us to collect essentially all the sections throughout an entire worm sample, which could then be aligned and stacked to generate a 3D model of spatial gene expression information (Fig. 2A). To improve tissue attachment to the Visium slides we pre-embedded with a methanol fixation step, generated 8 µm thick cryosections, placed multiple sections on the same Visium capture area without overlapping the OCT, and modified the H&E staining to be performed inside slide cassettes with intermittent warming. To attain a finer image for morphological annotation, we introduced imaging with z-stacks. Overall, the implementation of these technical changes to the Visium protocol enabled us to obtain spatially resolved transcriptomic information from the worm specimens.

We then applied this newly developed miniature-ST technique to the tissues of selected regions of *B. malayi* adult female worms and generated a ST dataset from 3 worm samples (i.e., samples BM1, BM2, BM3) and across a total of 30 sections (Fig. 1A, Supplementary Fig. 1, Supplementary Data 1). In total, we captured 7724 unique *B. malayi* genes from the studied region containing ovary tissue, the beginning of the uterus with fertilized eggs and early embryos, as well as part of the digestive tract and body wall. These captured genes represent ~66% of the 11,777 *B. malayi* genes in the annotated genome (WormBase: WBPS14[40]), across 547 spots with an average of ~1457 unique molecules (UMIs) per spot and ~529 unique genes per capture spot. The number of genes we captured is in a similar range to the 8000-10,000 genes (70-90% of genome) per sequence library captured by a previous bulk RNAseq study across whole worms and different life-cycle stages[41]. We observed similar unique gene and UMI distributions per spot across the different samples (Fig. 1B) and across sections from the same sample (Fig. 1C). In addition, different samples ($r = 0.86-0.97$, $p < 0.05$) and different sections from the same sample ($r = 0.87-0.98$,

$p < 0.05$) had a high correlation in their average gene expression (Fig. 1D). Such results demonstrate that spatial transcriptomic information was reproducibly captured across samples and sample sections with our miniature-ST method.

## Spatially distinct gene expression separation of digestive tract, body wall, and multiple reproductive tissue regions

Given the high reproducibility of the approach, we then performed unsupervised clustering analysis of the miniature-ST data and identified four distinct clusters (Fig. 2B, C). Differential expression analysis identified differentially expressed (DE) genes for each cluster, with 36 DE genes in cluster 1, 82 DE genes in cluster 2, 58 DE genes in cluster 3, and 31 DE genes in cluster 4 that significantly changed their expression in each cluster as compared to the other 3 clusters (Supplementary Data 2A). The significantly upregulated DE genes (positive logFC, $p < 0.05$) in each cluster were annotated using a previous proteomics study[25], which identified each cluster as specifically enriched in a set of highly expressed tissue-specific markers (Methods). Thus, we considered each cluster as representative of the tissue type for which the cluster contained highly expressed tissue specific markers: digestive tract for cluster 1, reproductive tissue for clusters 2 and 4, and body wall for cluster 3 (Fig. 2D, Supplementary Data 2B–F). Of note, cluster 3 was a mixed cluster with higher enrichment in body wall marker genes (9.3%), but also contained markers for the reproductive tract (4.7%) (Fig. 2D, Supplementary Data 2B, C). When looking at the spatial distribution of cluster 3 on the tissue sections (Fig. 2C), we observed that the spots localized to both tissue types identified at the gene expression level: the body wall (hypodermal cord, muscles) found towards the exterior part of the worms, and the reproductive tissue located inside the worms. Thus, with a resolution of 55 µm, we observed at both the spatial spot localization level and at the gene expression level a tissue-specific separation by clustering.

To study how gene expression patterns localize to different morphological structures and to see if there are specific patterns that arise across the entire worm region, we explored different cluster marker gene spatial distributions in 3D by compiling all the consecutive sections from worm sample BM2 (Fig. 2A). In addition, we performed fixed-term enrichment analysis to provide an overview of the genes and processes enriched in each cluster (Fig. 2E, Supplementary Data 3A-B). By visualizing the expression patterns of marker genes for the digestive tract, the reproductive tract, or the body wall (Fig. 3A), we could localize the corresponding tissue region in the 3D reconstruction (Fig. 3B-F). Specifically, cluster 1 contained DE genes associated with the cell surface and interactions between cells and the environment. The most significant DE genes for cluster 1 were ones with ribosomal function and its components, ubiquitin functions, and Transthyretin-like family proteins ($p < 0.001$) (Fig. 2E, Supplementary Data 3A-B). Transthyretin is a protein that transports the thyroid hormone thyroxine and Vitamin A (retinol)[42] and proteins annotated as Transthyretin-like are associated with the cell surface in *B. malayi*. Moreover, transthyretin-like proteins were previously identified as secretory *B. malayi* proteins with immunomodulatory potential[43], confirming their role in parasite-host interactions. Transthyretin-like protein families are potential vaccine candidates against human filariasis[44]. In terms of interactions between cells and the environment, we observed a positive expression of genes encoding proteins involved in ion binding activity, ion transmembrane transporter activity, sterol-binding and transfer activity, and ShKT-domain containing proteins (Supplementary Data 3A-B). The largest family of proteins containing a ShKT-line domain is found in worms *Caenorhabditis elegans*, *C. briggsae*, *B. malayi*, *B. pahangi*, *Ancylostoma ceylanicum*, *S. mansoni*, and *Toxocara canis*[45,46]. ShKT-domain containing proteins from parasitic nematodes were recently shown to possess immunomodulatory activity via the blockage of voltage-gated potassium channels on human effector memory T cells[45]. It was also

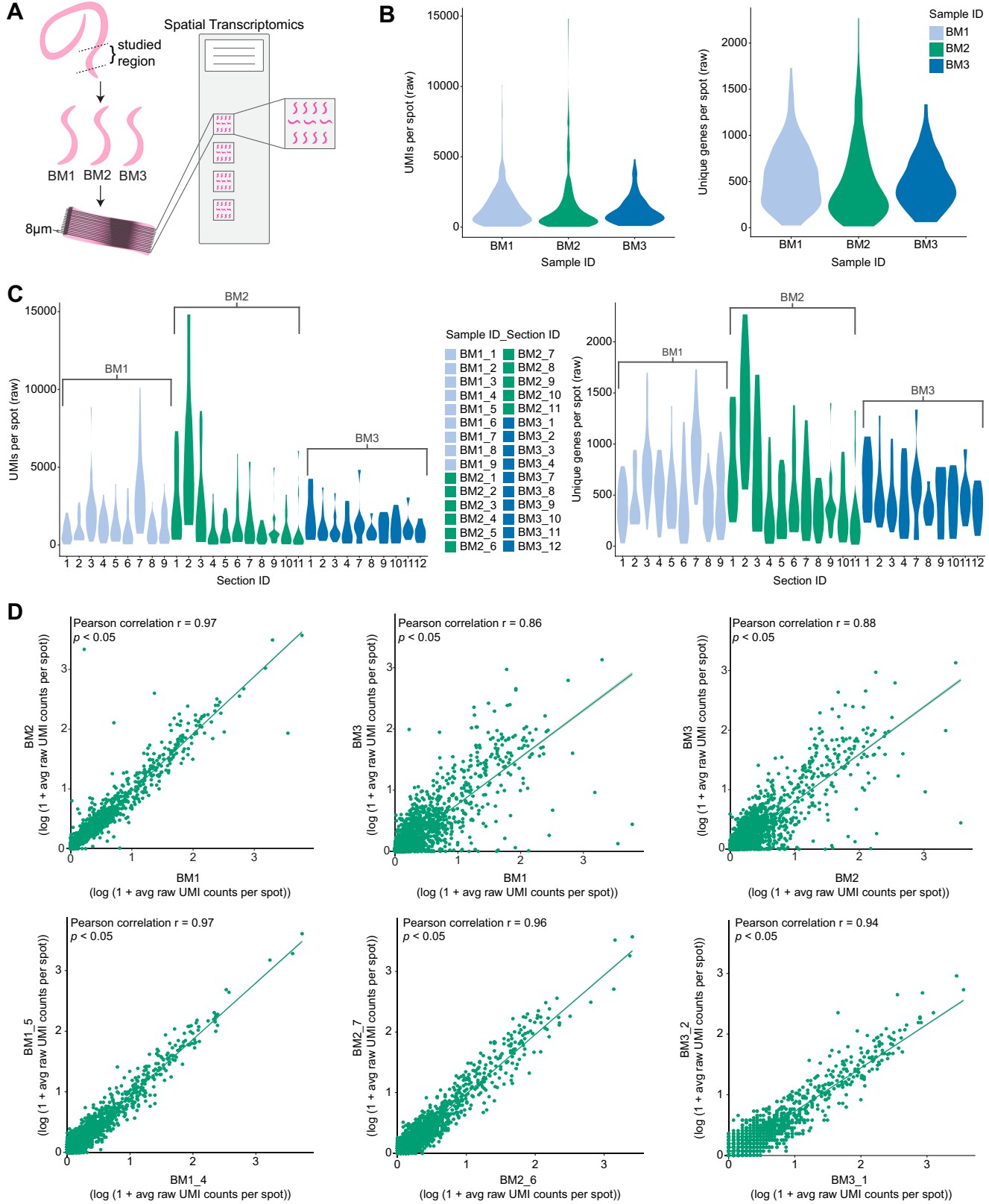

**Fig. 1 | Reproducibility of miniature-ST. A** Overview of the miniature-ST method performed in this study, where cryosections from the posterior region of adult female *Brugia malayi* worms containing ovary tissue, the beginning of the uterus with fertilized eggs and early embryos, the digestive tract, and body wall were analyzed using Spatial Transcriptomics (ST). **B** Violin plots of the unique molecules (UMIs) and unique genes per spot across the different worm samples used in the study. **C** Violin plots of the UMIs per spot and unique genes per spot across different worm sample sections (sample name_consecutive section #) used in the study. **D** Scatter plots of the log(1 + average gene expression) between worm samples (top, $r = 0.86\text{-}0.97$, $p < 0.05$) and sections from the same sample (bottom, $r = 0.94\text{-}0.97$, $p < 0.05$) across all genes. The statistical test is based on the Pearson's product moment correlation coefficient (r) with 95% confidence intervals and $p$ values ($p$) were estimated with a two-sided alternative hypothesis.

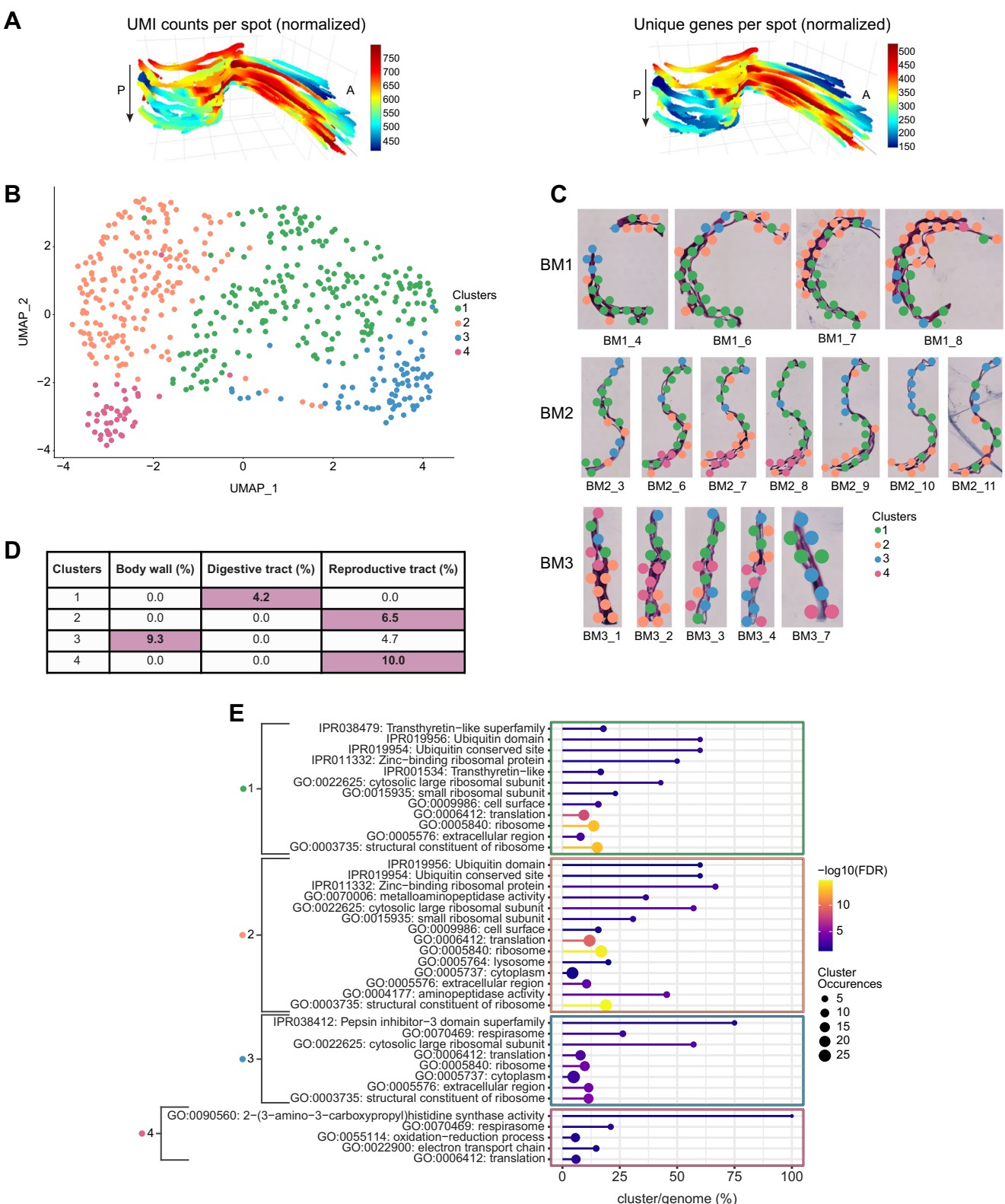

**Fig. 2 | Tissue region specific spatial gene expression patterns. A** 3D model of the UMIs and unique genes per spot across worm sample BM2. Magnification 20x. A: Anterior, P: Posterior, arrow indicates first to last section through the worm. **B** UMAP showing the four clusters of *Brugia malayi* transcriptome data. **C** Spatial distribution of the four clusters on tissue sections from *B. malayi* samples.

Magnification 20x. Sample names are to the left of each row of section images and section names are below each section image. **D** Percent of differentially expressed marker genes per cluster enriched for specific tissue regions. **E** Significantly over-represented functional terms for each cluster were identified using a two-sided Fisher's exact test (FDR < 0.05).

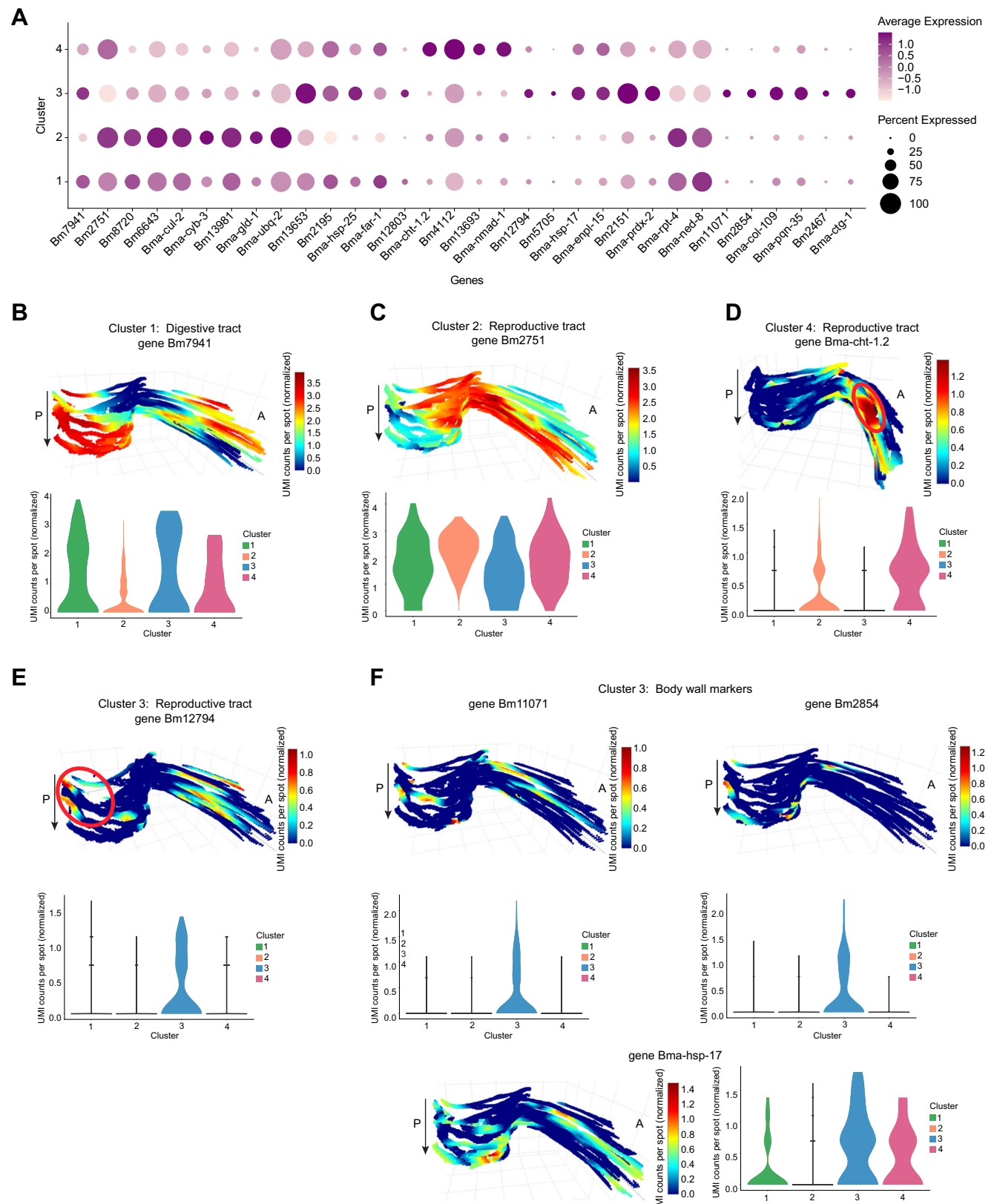

**Fig. 3 | Tissue region expression of cluster marker genes. A** Dotplot depicting the expression of *B. malayi* cluster marker genes across the different clusters. Average expression values indicate the average UMI counts (normalized) per spot across all spots in each cluster. Percent expressed values indicate the percentage of spots within each cluster that express each gene. **B–F** Spatial expression of cluster marker genes in 3D through worm sample BM2 and expression distribution across each cluster in violin plot. Magnification 20x. A: Anterior, P: Posterior, arrow indicates first to last section through the worm.

shown that some ShKT-like domain-containing proteins were highly expressed along the digestive tract in *C. elegans* adult worms[47]. Additionally, the expression of a marker for the digestive tract (cluster 1), Bm7941, changed throughout the entire 3D worm sample. Specifically, when moving through the 3D model of the worm sample, the elevated expression of Bm7941 shifted from the anterior side to the posterior side, further supporting cluster 1 as localizing to the digestive tract (Fig. 3B, Supplementary Data 2D).

According to the proteomic analysis study[25] used to annotate our cluster DE genes, three genes (Bm2751, Bm8720, Bm6643) upregulated in cluster 2 are markers for reproductive tissue (Fig. 3A, Supplementary Data 2A,E). The expression of these genes in cluster 2 was elevated in the middle, closer to the posterior region, and then shifted towards the anterior part of the worm through the 3D model (Fig. 3C, Supplementary Data 2E). Cluster 2 covers a substantial internal part of the worm and, according to highly expressed reproductive tissue markers, represents a tissue region containing the ovaries and the uterus. We observed that more than 13% of all up-regulated genes in this cluster are involved in processes associated with oogenesis and early embryogenesis (Supplementary Data 2A). According to their *C. elegans* orthologs, these genes are part of the following processes: meiotic chromosome segregation and organization (Bma-cul-2, Bma-cyb-3), oocyte maturation (Bma-cyb-3), gamete generation (Bm13981), positive regulation of female gonad development (Bma-gld-1), and polarity specification of the anterior-posterior axis (Bma-ubq-2) (Fig. 3A, Supplementary Data 2A). This pattern confirms that cluster 2 truly represents the reproductive tissue of worms. Down-regulated genes in cluster 2 consist of heat-shock proteins (Bm13653, Bm2195, Bma-hsp-25), and lipid binding and lipid droplet disassembly processes (Bma-far-1, Bm12803) (Fig. 3A, Supplementary Data 2A). During oogenesis, lipids are stored for later use in rapid, early embryogenesis[48]. Therefore, the down regulation of genes involved in the degradation of stored lipids indicates that cluster 2 is closer to the ovaries than the uterus region of the reproductive tract.

Clusters 3 and 4 also contained markers for the reproductive tract, however, these clusters appeared to localize to different reproductive tract tissue regions than cluster 2. The expression of differentially expressed marker genes Bma-cht-1.2 and Bm4112 for cluster 4 partially overlapped the cluster 2 reproductive tract markers expression pattern; clusters 2 and 4 were also located closer to one another both in UMAP space and spatially on the tissue sections (Fig. 2B, C, Fig. 3C, D). However, the cluster 4 markers could represent a different reproductive tract region than cluster 2 (Fig. 3A, C, D, Supplementary Data 2A, F). After oocytes are fertilized in the adult female worm reproductive tract, embryos build a shell mainly composed of chitin and develop into microfilaria (a pre-larval worm stage) within this chitin shell. Bma-cht-1.2 is involved in chitin binding activity and thus the region where Bma-cht-1.2 showed elevated expression may represent the viaduct region of the reproductive tract, which connects the ovaries to the uterus; the viaduct is where fertilization and very early formation of the embryos occur (Fig. 3D, Supplementary Data 2F). In addition, cluster 4 displayed a significant enrichment of genes encoding proteins with respiration activity, including the electron transport chain, respirasome and oxidative-reduction process, histone (H4), and histone binding protein synthesis (Fig. 2E, Supplementary Data 3A, B). Histones and histone-binding proteins, such as those encoded by Bm4112 and Bm13693, are important for developing embryos in the reproductive tract, which corresponds to the region used in this study. The histone H4 (Bm4112) upregulated in cluster 4 ($p = 0.0004$) is an ortholog of *C. elegans his-10, his-31*, and *his-64* (Fig. 3A, Supplementary Data 2F). In *C. elegans, his-10* (a histone H4 gene) is responsible for chromatin formation and involved in the defense response to Gram-negative bacteria and the innate immune response[49,50]. In cluster 4, we also observed higher expression of genes essential in oogenesis, such as Bma-nmad-1, an ortholog of *C. elegans nmad-1* (Fig. 3A, Supplementary

Data 2A). In *C. elegans, nmad-1* is involved in meiotic chromosome condensation, positive regulation of organelle organization, and positive regulation of oviposition[51]. Due to the upregulation of genes involved in oogenesis and fast-dividing embryos in the early steps of embryogenesis, we concluded that cluster 4 represents the region of the uterus with fast dividing cells of early embryos and this region is adjacent via the viaduct to the ovaries (cluster 2).

Cluster 2 likely represents the ovarian part of the reproductive tract, cluster 4 the viaduct and uterus portions of the reproductive tract, while cluster 3 contains differentially expressed markers (Bm12794, Bm5705) that represent a third, distinct region of the reproductive tract (Fig. 3A, Supplementary Data 2C). Cluster 3 markers Bm12794 and Bm5705 show elevated expression in the posterior part of the worm, a largely distinct pattern from the central, ovarian (cluster 2) and viaduct (cluster 4) reproductive tract marker expression patterns (Fig. 3E). The cluster 3 reproductive tract region could correspond to the uterus. Like cluster 1 (digestive tract), cluster 3 includes a substantial set of upregulated heat-shock proteins (Bm13653, Bma-hsp-17, Bma-hsp-25) (Fig. 3A, Supplementary Data 2A). Activation of heat shock proteins in the cluster correlated with an upregulation of unfolded protein binding activity (Bma-enpl-1) and peroxidase activity (Bm2151, Bma-prdx-2). However, proteasome-mediated protein catabolic activity (genes Bma-rpt-4 and Bma-cul-2) is downregulated in this cluster. Finally, cluster 3 showed a downregulation of Bma-ned-8 (Fig. 3A, Supplementary Data 2A). Likely active stress-related processes found in cluster 3 result in the downregulation of Bma-ned-8. In *C. elegans, ned-8* is involved in the regulation of the apoptotic process through signal transduction by a p53 class mediator[52–54]. As cluster 3 also has markers of the reproductive tract, we assumed that *ned-8* is more involved in this tissue. It was shown that elimination of *w*Bm from *B. malayi* initiates programmed cell death in the germline in ovaries and in the embryos in the uterus[10]. The induction of apoptosis was determined through increased expression of the cell death protein-3 (*ced-3*) gene, and the increase in the amount of inactive and active (cleaved) CED-3 protein forms in antibiotic-treated *B. malayi* worms as compared to untreated (control) worms[10]. In addition, cluster 3 had 2x the number of markers for the body wall (Bm11071, Bm2854, Bma-hsp-17) than markers for the reproductive tract (Fig. 2D, Fig. 3A, Supplementary Data 2B, C), where the body wall consists of muscle, cuticle, hypodermal cells, and nerve cells. These body wall markers (Bm11071, Bm2854, Bma-hsp-17) showed elevated expression in the outer areas of the worm in the first and last sections of the stack (Fig. 3F), an expected expression pattern considering the first and last sections will likely contain the outermost portions of body wall when collecting longitudinal sections. Furthermore, Bm11071 and Bm2854 encode predicted cuticle structural constituent proteins as the cuticle forms the outermost region of the body wall, their expression pattern further supports cluster 3 as partially localizing to the body wall region of the worm. In addition to the overexpression of genes that are structural constituents of the cuticle (Bm2854, Bma-col-109, Bm11071), Bma-pqn-35 (Bm10058), a gene ortholog of a *C. elegans* gene and, according to its annotation, expressed in muscle cells, was also overexpressed in cluster 3 (Fig. 3A, Supplementary Data 2A). The upregulation of genes encoding cuticle and muscle-related genes further supports the partial localization of cluster 3 to the body wall tissue region.

We observed an elevation of expression of Bm2467 and Bma-ctg-1 genes in cluster 3, indicating the increase of fatty acid metabolic processes in the body wall region of the parasite as Bm2467 is predicted to enable 3-hydroxyacyl-CoA dehydrogenase activity and enoyl-CoA hydratase activity, and Bma-ctg-1 is a lipid binding protein (Fig. 3A, Supplementary Data 2A). As the body wall consists of high metabolic tissues, such as muscle and hypodermal cells, the higher activity of metabolic processes is expected in these regions. Interestingly, we showed that metabolic processes for lipids/fatty acids turnover were

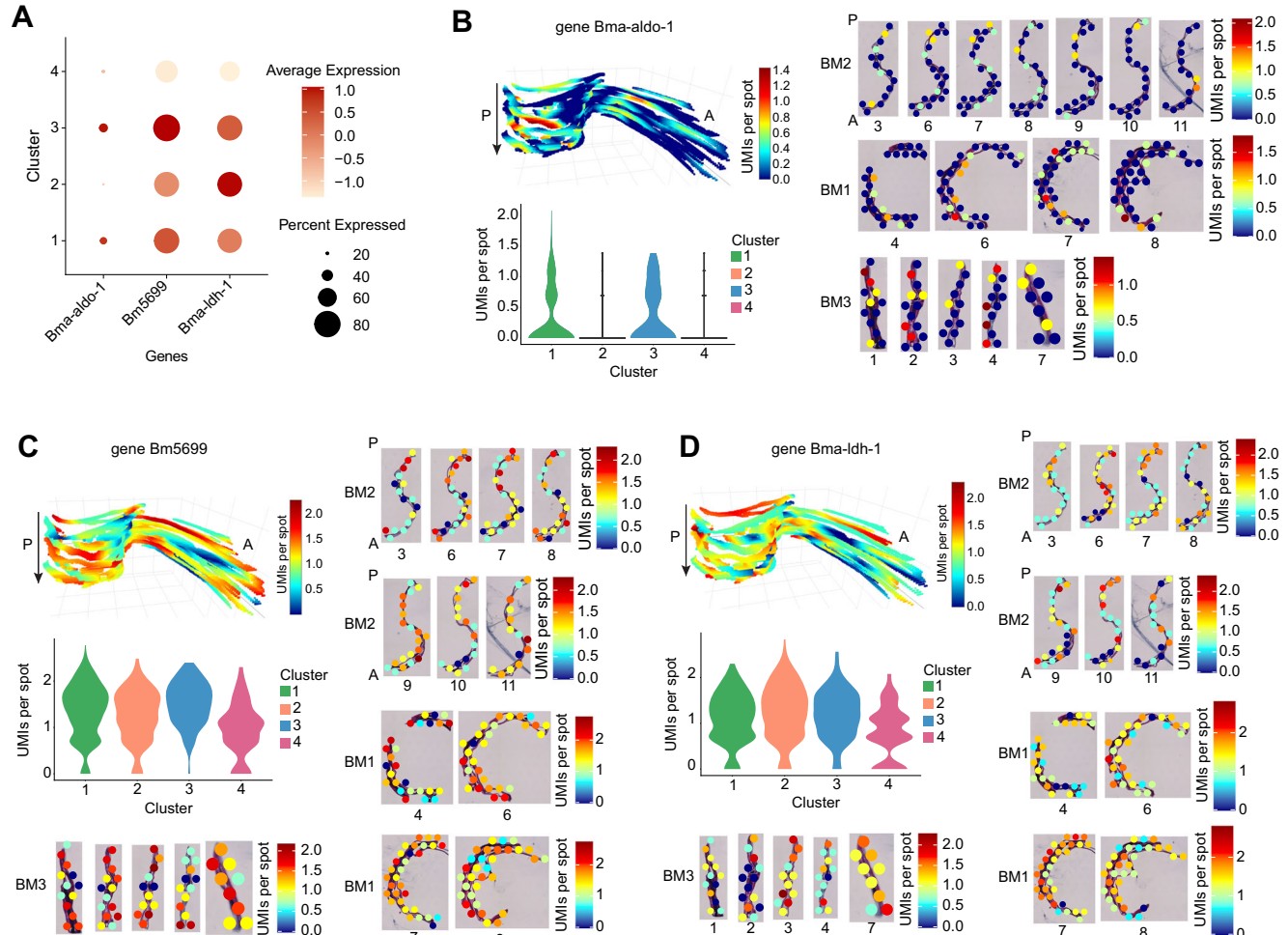

**Fig. 4 | Spatial distribution of *B. malayi* glycolytic enzyme genes. A** Dotplot depicting expression of glycolytic genes across the different clusters. Average expression values indicate the average UMI counts (normalized) per spot across all spots in each cluster. Percent expressed values indicate the percentage of spots within each cluster that express each gene. **B**–**D** 3D model of gene expression in sample BM2, 2D gene expression across select sections from all 3 samples (normalized UMI counts per spot), and the gene expression distribution (normalized UMI counts per spot) across the clusters in a violin plot for glycolytic genes Bma-aldo-1 (**B**), Bm5699 (**C**), and Bma-ldh-1 (**D**). Magnification 20x. A: Anterior, P: Posterior, arrow in each 3D plot indicates first to last section through the worm. Sample names are to the left of each row of section images and section numbers are below each section image.

the most significant for the body wall tissues of *B. malayi* (Fig. 3A, Supplementary Data 2A). Recently, it was shown that *Wolbachia* residing in the hypodermal cells of the parasites induce and use the glycolytic pathways[19,55]. We hypothesize that fatty acid metabolism may also be involved in providing carbohydrates to symbiotic bacteria. Overall, the clustering analysis facilitated an exploration of nuanced spatial gene expression patterns, as we could spatially pinpoint the expression of genes and pathways that were differentially regulated in the digestive tract, the body wall, and multiple reproductive tract tissue regions within intact, longitudinal tissue sections.

**Spatial localization of key *B. malayi* glycolytic enzyme genes**

We next focused specifically on the spatial expression patterns of key glycolytic enzyme-related genes that are important in the *B. malayi* - *Wolbachia* mutualistic relationship. Pyruvate is one of the most essential metabolites for prokaryotic cells. *Wolbachia* has the full complement of genes that can use pyruvate for gluconeogenesis and for energy metabolism via the tricarboxylic acid cycle (TCA cycle)[55]. However, *Wolbachia* is missing some key enzymes that are needed for making pyruvate[19]. In previous studies, we showed that glycolysis and other pathways that produce pyruvate in filarial worms (such as *B. malayi*) provide pyruvate to *Wolbachia*[19,55]. Thus, glycolytic enzymes were shown to play an important role in maintaining the mutualistic

association between *w*Bm and *Brugia* worms. Therefore, we analyzed the genes involved in *B. malayi* pyruvate metabolism between different tissues (clusters) of the worm. We hypothesized that we could define the source and location of the pyruvate used by *Wolbachia* within *B. malayi*. We looked specifically for genes associated with glycolysis (the pathway that produces pyruvate), gluconeogenesis (the pathway where pyruvate is used to synthesize glucose, and precursors for nucleotide biosynthesis), lactate dehydrogenase (where lactate is converted to pyruvate and the reverse), and enzymes that convert cysteine amino acids to pyruvate (Supplementary Data 4).

Although our miniature-ST method captured many glycolytic genes that can be explored in our shiny app (https://giacomellolabst. shinyapps.io/brugiast-shiny/), we focused on three genes (Bma-aldo-1, Bm5699, Bma-ldh-1) found among our cluster DE genes (Fig. 4A). Bma-aldo-1 (aldolase-1) and Bm5699 (glyceraldehyde 3-phosphate dehydrogenase) encode an enzyme for glycolysis and showed increased expression in the digestive tract (cluster 1) and body wall (cluster 3) clusters but were downregulated in the reproductive tract clusters (clusters 2 and 4) (Fig. 4A-C, Supplementary Data 2A). *Wolbachia* are located in the hypodermal cells (part of the body wall) and in the ovaries, oocytes, and developing embryos within the uterus. However, *w*Bm is most abundant in the lateral cords (part of the body wall) as compared to the oocytes and embryos that contain few *w*Bm. We

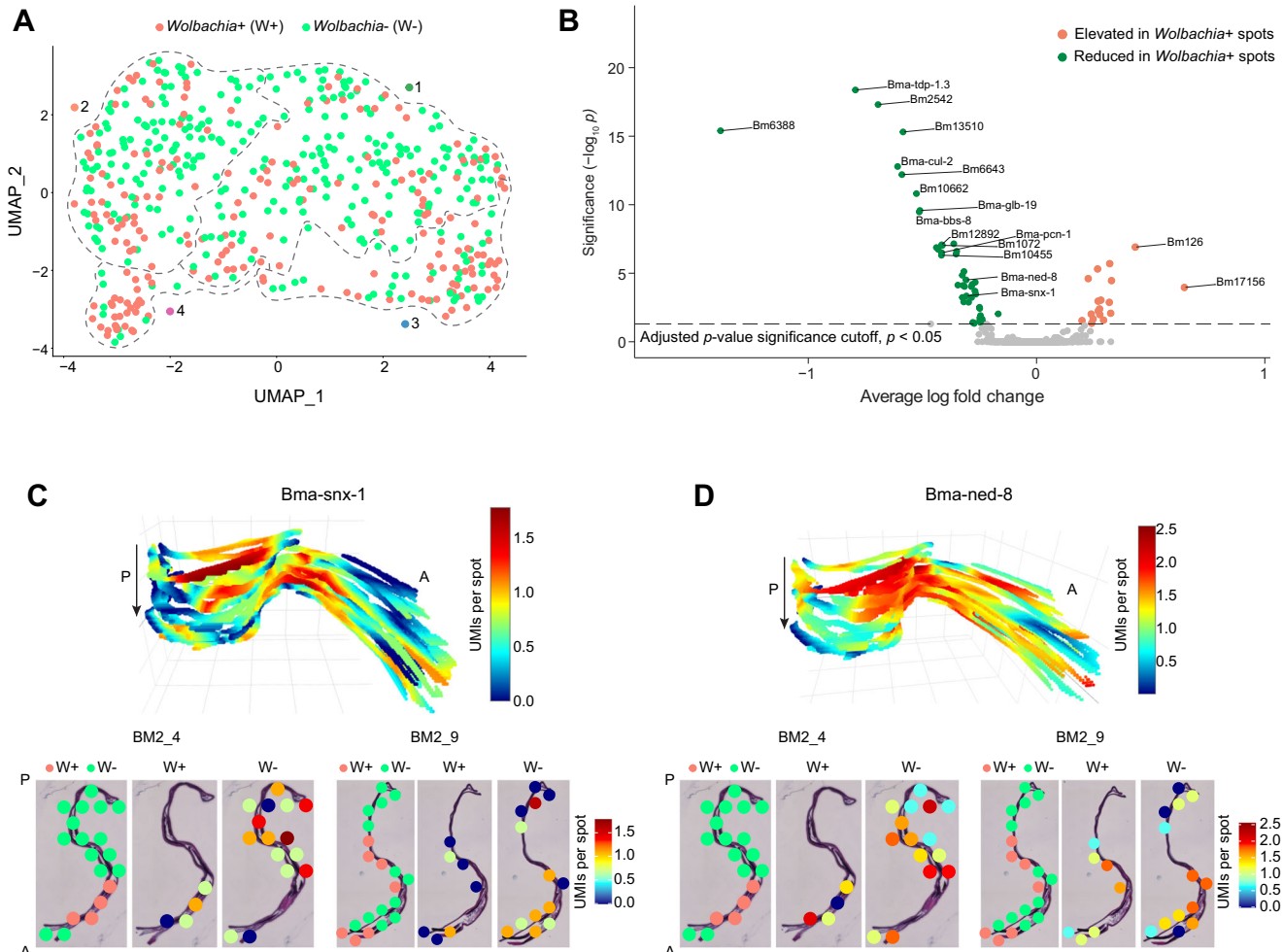

**Fig. 5 | Co-localization of *B. malayi* genes in *Wolbachia*+ versus *Wolbachia*-spots. A** UMAP showing spots with presence (*Wolbachia* + /W + ) or absence (*Wolbachia*-/W-) of *Wolbachia*. Dotted lines outline the different clusters from Fig. 2B. **B** Volcano plot showing the differentially expressed genes in *Wolbachia*+ versus *Wolbachia*- spots (green: average log fold change <0.4 and adjusted *p* value < 0.05, orange: average log fold change > 0.4 and adjusted *p*-value < 0.05).

Differential expression analysis used the Wilcoxon Rank Sum test and adjusted *p* values (*p*) were estimated with a two-sided alternative hypothesis. **C**–**D** 3D model and 2D images of gene expression in sample BM2 (sections BM2_4 and BM2_9) for two differentially expressed genes, Bma-snx-1 (**C**) and Bma-ned-8 (**D**), between *Wolbachia* + (W + ) and *Wolbachia*- (W-) spots. Magnification 20x. A: Anterior, P: Posterior, arrow indicates first to last section through the worm.

suspect that the presence of *Wolbachia* in the body wall, where the bacterial load is at its highest, could induce the expression of glycolytic enzymes, resulting in the production of pyruvate and thus ensuring bacterial survival; this is not the case in the reproductive tissues[16,19,55]. The third gene, Bma-ldh-1 (lactate dehydrogenase or LDH) showed elevated expression in cluster 2 (reproductive tract) and was down-regulated in cluster 3 (reproductive tract and body wall) (Fig. 4A, D, Supplementary Data 2A). LDH converts lactate to pyruvate and pyruvate to lactate, which may be a reason for the differential expression of this enzyme between clusters 1 and 3. Taken together, we could spatially localize key *B. malayi* - *w*Bm genes within tissues. We hypothesize that *Wolbachia*-induced glycolytic enzyme production occurring in the *B. malayi* body wall produces the pyruvate on which the symbiont depends.

### Co-localization of *B. malayi* genes in *Wolbachia*+ compared to *Wolbachia*- regions of the tissue

When capturing polyadenylated transcripts using the miniature-ST method, we also non-specifically captured *Wolbachia* tRNA and rRNA and inferred that this signal would be indicative of *w*Bm localization. We ran our miniature-ST library of all sequencing reads against the *Wolbachia* tRNA and rRNA gene sequences, as those are typically the most highly abundant transcripts in an organism (Methods). We detected at least 1 UMI from 33 tRNA/rRNA *Wolbachia* genes across 206 spots (39.5% of spots) in the worm samples (Fig. 5A). This allowed us to colocalize *B. malayi* gene expression in relation to *Wolbachia* by determining the *B. malayi* differentially expressed (DE) genes between spots containing *Wolbachia* (*Wolbachia* + ) and spots lacking *Wolbachia* (*Wolbachia*-) in the worms. This approach identified 65 DE parasite genes (*p* < 0.05) with 48 downregulated genes and 17 upregulated genes (Fig. 5B, Supplementary Data 5). The presence of *Wolbachia* in the tissue (determined by detecting tRNA/rRNA bacterial transcripts at that location) is associated with a significant downregulation in the expression of *B. malayi* genes that encode proteins annotated with functions that include protease and peptidase activity, endopeptidase activity, and proteasome-mediated protein catabolic process (Supplementary Data 5). Suppression of protease activity can potentially reduce the proteolysis in *Wolbachia*+ tissue of female worms. We hypothesized that the inhibition of proteolysis in *Wolbachia*+ tissue protects bacterial proteins, including those that are involved in the interaction with the host.

We noticed that the relative abundance of *Wolbachia* (in terms of UMIs per spot) varied across the *B. malayi* samples, sections, and clusters (Supplementary Fig. 2A–C). The number of *Wolbachia* is

higher in the hypodermal cells of the body wall than in oocytes (ovary) and developing embryos (uterus)[56], and correspondingly, we observed higher levels of *Wolbachia* UMIs per spots in cluster 3 (annotated as the body wall) (Supplementary Fig. 2C). The same cluster also showed increased expression of the previously discussed glycolytic enzyme genes (see "Spatial localization of key *B. malayi* glycolytic enzyme genes", Fig. 4A–C). Despite these observations, glycolytic enzyme genes were not differentially expressed between *Wolbachia+* and *Wolbachia-* spots. There was, however, a correlation between the expression of several glycolytic enzyme genes (Bma-enol-1, Bma-hxk-1.1, Bm9363) and *Wolbachia* abundance (Pattern 3 in Supplementary Fig. 2D, Supplementary Data 6C). Specifically, these co-expressed genes (Bma-enol-1, Bma-hxk-1.1, Bm9363) were upregulated in spots with higher levels of *Wolbachia* (Pattern 3 in Supplementary Fig. 2D, Supplementary Data 6C), aligning with previous studies showing that *B. malayi* glycolytic enzymes are strictly dependent on *Wolbachia* activity as *B. malayi* provides pyruvate to the symbiont that is then used for the production of energy and nucleotides[19,55]. In addition, across the *B. malayi* genes correlating with *Wolbachia* abundance, we observed an enrichment of Gene Ontology (GO) terms related to stress reactions, including ROS, oxidative stress, and response to cellular toxins (Supplementary Fig. 2E, Supplementary Data 7C). We hypothesize that the bacteria may be responsible for initiating these reactions in the host cells.

Overall, our analyses revealed a general response of the *B. malayi* cells to their *Wolbachia* endosymbiont. For example, across the downregulated genes in *Wolbachia+* spots, two genes caught our attention: Bma-snx-1 and Bma-ned-8 (Fig. 5B-D, Supplementary Data 5). Bma-snx-1 is an ortholog of *C. elegans snx-1*, and it is predicted to enable phosphatidylinositol binding activity (SNX). SNX proteins promote phagosome-lysosome fusion and are involved in apoptosis and autophagy-mediated elimination of apoptotic cells[57,58]. We hypothesized that the inhibition of SNX proteins in *Wolbachia+* tissue indicates that *Wolbachia* could be suppressing the late steps of autophagy and endosomal/phagosomal degradation by blocking their fusion with lysosomes. *Wolbachia* itself is surrounded by a host-derived vacuole and can be recognized as a phagosome in the cytoplasm of host cells[59]. Therefore, we predicted that SNX proteins could play a role in protecting *Wolbachia* in the cytoplasm of eukaryotic host cells. Bma-ned-8, an ortholog of *C. elegans ned-8*, contains ubiquitin-like domains and is also involved in the regulation of the apoptotic process during embryogenesis of *C. elegans* worms[53]. As these genes are downregulated in tissues with *Wolbachia*, we hypothesized that the presence of *Wolbachia* can decrease apoptotic processes in developing embryos and that *ned-8* plays a key role in this regulation (Fig. 5B-D, Supplementary Data 5). It is known that the elimination of *Wolbachia* increases intensive apoptosis in germ cells and in the developing embryos of antibiotic-treated female worms (7 days treatment)[10]. However, we reasoned that since the elimination of the bacteria reduces the support of the worm's biological processes, consequently it could induce programmed cell death in the reproductive system of the worms. It is also possible that the elimination of *Wolbachia* has a direct effect on the expression of genes that regulate apoptotic processes in *B. malayi* worms. Further experimental work is needed to validate these hypothesized relationships between *B. malayi* and *Wolbachia*. However, our exploratory approach localizes gene-specific interactions within different spatial compartments across tissue regions containing or lacking *Wolbachia*.

### Spatial transcriptomic patterns in post-doxycycline treated worms

To understand the impact of chemotherapeutic treatment on spatial gene expression patterns in the selected region of the adult *B. malayi* worm, we treated worms with doxycycline (12.5 μM, in vitro 3 days, see Methods) and obtained preliminary ST data from 5 sections across 2 treated worm samples (T1 and T2) (Supplementary Data 8). We

observed similar unique molecule (UMI) and unique gene distributions per spot across the sections and samples, with a high correlation of average gene expression between sections ($r = 0.82$-$0.94$, $p < 0.05$) (Fig. 6A-B). Doxycycline was previously observed to reduce the number of *Wolbachia* present in adult female worms, inducing apoptosis in developing embryos of treated females and significantly decreasing the amount of microfilariae released by treated females[7,59–62]. Accordingly, we observed a decrease in *Wolbachia* transcripts (UMIs) in the treated worms compared to control worms, with 15.5% of spots annotated as *Wolbachia+* spots in the treated worms (Fig. 6C). Both control and treated worms showed a higher distribution of UMIs and unique genes per spot in *Wolbachia+* spots compared to *Wolbachia-* spots (Fig. 6D). Worms under doxycycline treatment showed lower *Wolbachia* UMI and unique gene distributions per spot compared to control worms in both *Wolbachia+* and *Wolbachia-* spots (Fig. 6D), pointing to globally reduced gene expression levels after 3 days of doxycycline treatment.

To explore the impact of doxycycline treatment on the expression of *B. malayi* genes hypothesized to co-localize with *Wolbachia* (the 65 DE genes between *Wolbachia+* and *Wolbachia-* spots in the control worms), we compared gene expression across *Wolbachia+* and *Wolbachia-* spots in control versus treated worms (Fig. 6E, Supplementary Data 9). Almost all the genes of interest showed lower average expression in the treated worms, such as the previously discussed Bma-snx-1 and Bma-ned-8 (Fig. 6E, Supplementary Data 9A). Of the 65 DE genes (in control samples between *Wolbachia+* and *Wolbachia-*spot), 10 genes (Bma-cul-2, Bm13981, Bm10455, Bm12892, Bma-try-1, Bm14034, Bma-cyb-3, Bma-ubq-2, Bm1061, Bma-lap-2) were annotated to have protease activity (Fig. 6E-F, Supplementary Data 5). Two (Bma-cul-2 and Bm14034) of the 10 genes showed elevated expression in *Wolbachia+* spots compared to *Wolbachia-* spots in treated samples (Fig. 6E, Supplementary Data 9B). Bma-cul-2 and Bm14034 are likely involved in the degradation of dying *Wolbachia* in treated samples. Interestingly, parasite heme-binding proteins (Bm126, Bma-glb-19) that are part of *Wolbachia*-host symbiosis[14], were downregulated in treated samples compared to controls (Fig. 6E, F, Supplementary Data 9A). However, Bm126 is still elevated in *Wolbachia+* spots compared to *Wolbachia-* spots in the treated samples (Fig. 6F, Supplementary Data 9B). This indicates that the short-term antibiotic treatment does not completely eliminate *Wolbachia* from the worms. Antibiotic treatment can also directly affect *B. malayi* gene expression separately from changes induced by the reduction of *Wolbachia*. We focused on looking at the expression changes in the treated worms among genes that were differentially expressed between *Wolbachia+* and *Wolbachia-* spots in the non-treated samples (Supplementary Data 5). The expression of some of these genes (Bm17156, Bm294, Bm16, Bma-bbs-8, Bma-rpl-33.1, Bm6111, Bm13981, Bm8720) also correlated with *Wolbachia* density in our non-treated samples (Supplementary Fig. 2D, Supplementary Data 6A-C), supporting the hypothesis that these genes showed transcriptional changes that colocalize with *Wolbachia*. Additional, comprehensive analyses that include spatial localization, treatment, and *Wolbachia* presence in a larger sample size could provide deeper insights into these tissue-specific effects under antibiotic treatment. Taken together, our results provide preliminary localization of gene-specific, doxycycline-induced effects on the *B. malayi - wBm* relationship.

## Discussion

In this work, we present miniature-ST, a method to analyze the spatial transcriptome of samples at the micrometer scale. We modified embedding, cryosectioning, fixation, and staining steps to enable the analysis of the spatial structures composing small pathogens, such as parasitic filarial worms, that cause a variety of infectious diseases. Our spatially-resolved characterization of the filarial parasitic nematode *Brugia malayi*'s transcriptome within intact, longitudinal sections, in

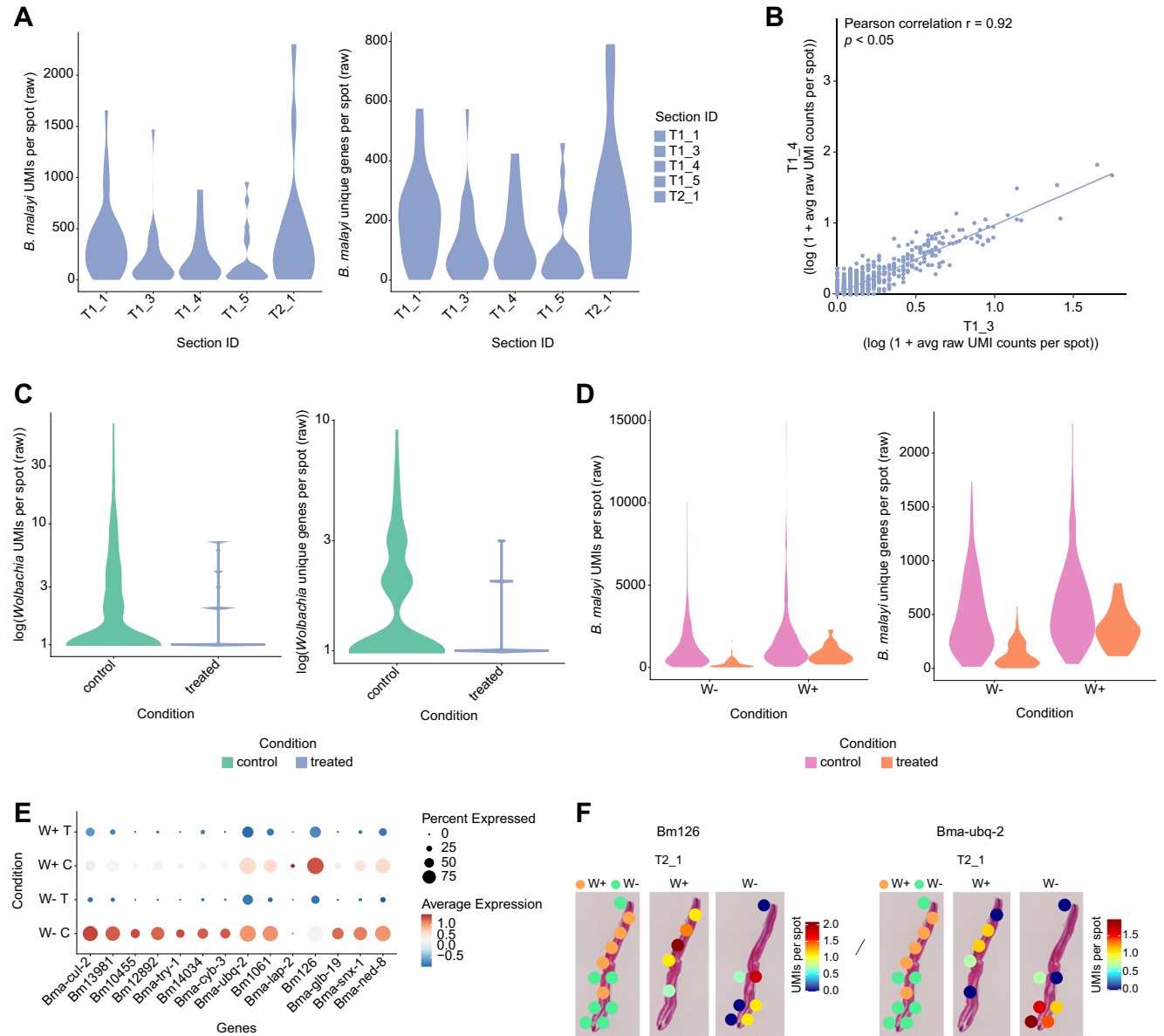

**Fig. 6 | Transcriptomic shifts under doxycycline treatment. A** Violin plots of the *B. malayi* UMIs and unique genes per capture spot across different treated worm tissue sample sections. **B** Scatter plot of the average *B. malayi* gene expression between consecutive treated worm sections (T1_4 and T1_5) across all genes (*r* = 0.92, *p* < 0.05). The statistical test is based on the Pearson's product moment correlation coefficient (r) with 95% confidence intervals and *p*-values (*p*) were estimated with two-sided alternative hypothesis. **C** Violin plots of the *Wolbachia* unique molecules (UMIs) per spot and unique genes per spot in control and treated worm sample sections. **D** Violin plots of the *B. malayi* unique molecules (UMIs) per spot and unique genes per spot in control and treated worm sample sections in

addition to the spatial capture of its endosymbiont *Wolbachia*, unveiled 2D and 3D tissue-specific gene expression patterns and the co-localization of *B. malayi* genes to specific tissue regions containing *Wolbachia*. In contrast to previous approaches[26–30] that analyzed transcript distributions across the depth of serial sections collected along the anterior-posterior (A/P) axis, or separated tissues with labor intensive dissection-based methods such as LCM[26,30], our miniature-ST method resolves structures and transcripts with 55 μm resolution within individual intact, longitudinal cryosections. This facilitates a 3D reconstruction of the analyzed area where transcripts can be explored along multiple axes (anterior-posterior, ventral-dorsal). Our study provides the additional advance of localizing the endosymbiotic

*Wolbachia* + (W + ) and *Wolbachia*- (W-) spots. **E** Dotplot depicting the expression of select *B. malayi* genes in *Wolbachia*+ spots in treated worms (W + _T), *Wolbachia* + spots in control worms (W + _C), *Wolbachia*- spots in treated worms (W-_T), and *Wolbachia*- spots in control worms (W-_C). Average expression values indicate the average UMI counts (normalized) per spot across all spots in each cluster. Percent expressed values indicate the percentage of spots within each cluster that express each gene. **F** 2D images of Bma-snx-1 and Bma-ned-8 gene expression in sample section T2_1 between *Wolbachia* + (W + ) and *Wolbachia*- (W-) spots. Magnification 20x.

bacteria, *Wolbachia*, within *B. malayi* to explore host-symbiont dynamics in a spatial context.

We applied our miniature-ST approach to visualize the spatial localization of *B. malayi* genes along a region in the adult female worms (i.e. one that contains ovary tissue, the beginning of the uterus with fertilized eggs and early embryos, part of the digestive tract, and body wall) in a series of 30 longitudinal cryo-sections across three different worms. We found that re-embedding the worm specimens onto flattened OCT improved the orientation of the sample block to the cryosectioning blade, facilitating the collection of higher quality sections. Worm samples in the original embedding (*n* = 1) and re-embedding (*n* = 2) showed consistent results in the Visium

experiments, and total RNA showed similar RNA quality between original and re-embedded worms across a larger sample size (duplicates, $n = 2$, for both originally embedded and re-embedded worms), confirming the re-embedding does not affect the gene expression information captured and rather improves the cryosection quality. We captured 66% of the genes in the annotated *B. malayi* genome with similar expression profiles across samples and sections, demonstrating the ability of the miniature-ST method to reproducibly capture a significant portion of the worm's transcriptome with spatial resolution. We identified an enriched set of tissue-specific markers—annotated according to a previous proteomics study[25]—in each cluster, indicating each cluster represented a distinct tissue type: digestive tract for cluster 1, reproductive tissue for clusters 2 and 4, and body wall for cluster 3. By constructing a 3D model of all consecutive sections throughout an entire worm sample, we observed that these tissue-specific gene expression patterns also shifted in the third dimension/z-plane along the ventral-dorsal axis and from anterior to posterior, through the worm region. Our study complements a spatial transcriptomic study of the *B. malayi* head region[26] by focusing on another area, a posterior region important in *B. malayi-wBm* symbiosis, generating a spatial dataset where gene expression can be explored throughout the region along multiple axes.

Furthermore, fixed term enrichment analysis revealed genes and processes enriched in each cluster/tissue. Specifically, the analysis showed the expression of genes associated with the cell surface, cuticle, and interactions between cells and the environment in cluster 1 for the body wall. Also, cluster 2 (annotated as reproductive tissue) contained up-regulated genes that are involved in processes associated with oogenesis and early embryogenesis. These findings confirm that the clusters represent specific tissues of the worms. Expression of marker genes for both clusters also shifted through the z-plane of the 3D model of the worm, further supporting the cluster's tissue-specific localization. For cluster 1 (body wall), significant DE genes included Transthyretin-like family proteins, where transthyretin-like proteins were previously identified as secretory *B. malayi* proteins with immunomodulatory potential[43] and as potential vaccine candidates against human filariasis[44]. In addition, we showed the spatial localization of key *B. malayi* glycolytic enzyme genes that are essential for worm-*Wolbachia* symbiosis[19,55]. For example, two glycolytic enzyme genes (Bma-aldo-1 (aldolase-1) and Bm5699 (glyceraldehyde 3-phosphate dehydrogenase)) were upregulated in *Wolbachia*-abundant tissue regions and downregulated in more *Wolbachia*-deficient tissue regions, which led us to hypothesize that *Wolbachia* may induce the expression of glycolytic enzymes. In addition, the expression of several other glycolytic genes (Bma-enol-1, Bma-hxk-1.1, Bm9363) correlated with *Wolbachia* abundance in our tissue sections. Increased expression of glycolytic enzymes results in the production of pyruvate, on which *Wolbachia* is dependent, and thus could be a mechanism to ensure bacterial survival. By spatially localizing key *B. malayi* - *w*Bm genes, we provide a new spatial, three-dimensional context to the symbiotic relationship. Miniature-ST facilitates the separation of different tissue regions, allowing better exploration of genes and pathways of interest within distinct spatial structures and across a 3D reconstruction of these parasitic worms.

To further explore the *B. malayi* - *w*Bm relationship, we identified regions of the tissue containing *Wolbachia* (*Wolbachia*+ spots), leveraging non-specific capture of *Wolbachia* transcripts by poly-d(T) capture, and explored how *B. malayi* genes are modulated in different tissue regions in the presence and absence of *Wolbachia*. Our co-localization analysis unveiled a potential reduction of proteolysis in the *Wolbachia*+ tissue of the female worms, protecting bacterial proteins from degradation and thus maintaining the bacterial interaction with the *B. malayi* host. Moreover, our data provided new insights into understanding the role of autophagy in the regulation of host-*Wolbachia* symbiosis[59]. Autophagy is a key host intracellular defense mechanism that regulates bacterial population abundance inside host cells. Co-localization analysis of *B. malayi* genes suggests that *Wolbachia* could be suppressing the late steps of autophagy and endosomal/phagosomal degradation by blocking phagosomal fusion with lysosomes. Specifically, expression of SNX proteins, which promote phagosomal fusion with lysosomes, autophagy, and apoptosis, was downregulated in *Wolbachia*+ spots, and thus could play a role in protecting *Wolbachia* in the cytoplasm of eukaryotic cells. We speculate that targeting these proteins could potentially be used to target the bacteria and therefore kill adult parasites. We further used miniature-ST to obtain preliminary data and evaluate the expression of *B. malayi* genes in response to anti-*Wolbachia* treatment. As expected, worms under doxycycline treatment had fewer *Wolbachia* and unique gene distributions per spot compared to control worms. Almost all of the 65 differentially expressed genes between *Wolbachia*+ and *Wolbachia*- spots in the control worms showed lower average expression in the treated worms compared to control worms, including the downregulation of heme-associated genes that are potentially involved in the mutualism between *Wolbachia* and *Brugia* parasites[63]. A limitation of using antibiotic treatment to deplete *Wolbachia* in *B. malayi* worms is that we were unable to completely separate the effects of *Wolbachia* reduction from the direct effects of doxycycline treatment on *B. malayi* gene expression. Future work could aim to give the worms more recovery time after doxycycline treatment to minimize its direct effects on *B. malayi*. Although our capture of *Wolbachia* was non-specific, we could spatially distinguish worm tissue regions containing *Wolbachia* and elucidate the regulation of *B. malayi* genes in those areas with miniature-ST. Our 3D model further refined the spatial distribution of the genes involved in these molecular processes, providing new, multidimensional spatial context to their expression in relation to *Wolbachia*. We provide a reference for transcriptome-wide transcript localization patterns across the studied region and within distinct tissue structures at 55 μm resolution, a resource not previously available at this scale. Future studies exploring the spatial co-localization of *B. malayi* gene expression in relation to *Wolbachia* could further address the symbiotic relationship of these two organisms. For example, additional experiments will involve studying antibiotic treated worms to further understand how antibiotic-induced *Wolbachia* clearance impacts gene expression of the worms. An ideal next step would be to directly identify *Wolbachia* genes in the worms to give a more nuanced picture of *Wolbachia* density in different *B. malayi* tissue areas and facilitate more in-depth co-localization studies, especially in response to antibiotic treatment.

The miniature-ST method could be extended to the full *B. malayi* adult worm or to other developmental stages. Additional studies could include parasitic worms within human tissue, providing a spatial exploration of the parasite actively infecting host tissue and opening the possibility of studying multilevel organism co-localization (*Wolbachia* endosymbionts inside the parasite and the parasite inside human host tissue). Limitations of the study include a spatial resolution of 55 μm, which does not yet allow single cell resolution. Imaging based spatial approaches offer the possibility of high, even subcellular, resolution; however, such methods focus on a specific set of target genes, making it difficult to design gene probes for species with a poorly annotated reference. ST, as a sequencing-based approach, is highly suitable for species that are not well annotated. Another study working with mouse embryos combined scRNAseq with a spatial transcriptomics approach[32]. Given that single cell isolation from *B. malayi* poses a challenge, potentially single nuclei RNAseq (snRNAseq) combined with miniature-ST could be used in the future to obtain single cell level information combined with the spatial context. An advantage of our miniature-ST approach is the ability to examine transcripts in difficult to isolate cells, such as the hypodermal syncytial cells that are as large as the worm's length, by working with intact tissue sections. Although working with small, micrometer scaled

samples poses technical challenges, an advantage is the possibility of collecting all sections through the entire worm to construct a 3D model of the gene expression patterns throughout the entire worm sample across all axes. Additionally, 3D reconstruction of spatial gene expression throughout the length of an entire worm, all tissue types included, would provide a wealth of information to explore. A 3D model not only confirms the localization of certain expression patterns to specific morphological regions, but also can provide new insights based on how these patterns shift throughout the entire specimen. Constructing a tissue atlas from small-scaled tissue is also economically advantageous, since all tissue sections can be collected for one sample on a single Visium capture area. The spatial gene expression information we collected across all sections from *B. malayi* is presented in a shiny app (https://giacomellolabst.shinyapps.io/brugiast-shiny/), offering the possibility for further exploration.

In conclusion, we present miniature-ST, a method to unlock the potential for exploratory studies of small infectious pathogens at a spatial scale. We provide the spatial characterization of *B. malayi* gene expression in a region of adult female worms, an important step in being able to spatially resolve micro-scale disease-causing pathogens that have their own spatial tissue structure. We not only spatially resolved the parasitic worm *B. malayi*, but also spatially localized its endosymbiotic bacteria, *Wolbachia*. This opens the way to exploring how *B. malayi* gene expression is spatially co-localized with *Wolbachia*. By spatially capturing the *B. malayi* transcriptome, this study forges a new path for studies that aim to spatially resolve gene expression information in other parasitic worms responsible for human infectious diseases. We envision our miniature-ST approach can be readily applied to other small pathogenic worms, providing new insights into the spatial organization of gene expression in these causative agents of a multitude of infectious diseases.

## Methods

### Ethics statement

All animal work conducted by the NIH/NIAID Filariasis Research Reagent Resource Center (FR3) followed the national and international guidelines outlined by the National Institutes of Health Office of Laboratory Animal Welfare and was approved by the University of Georgia Athens.

### Parasite material and treatment

*B. malayi* parasites recovered from the peritoneal cavity of infected gerbils (*Meriones unguiculatus*) were obtained from FR3, University of Georgia, Athens. Adult *B. malayi* female worms (>120 dpi) were placed in 7 ml of complete culture medium (RPMI-1640 supplemented with 10% FBS, 100 U/mL penicillin, 100 mg/mL streptomycin, 2 mM L-glutamine), 2 worms per well in 6-well plate, and incubated at 37 °C under 5% $CO_2$ conditions.

Control worms were incubated in the media for 2-3 days to assure they were alive, then worms were washed 3 times in PBS and fixed in 100% cold methanol (-20 °C) for 5 min. Treated worms were treated with doxycycline (12.5 uM) for 3 days in vitro, then the worms were immediately washed with PBS (3 times) and fixed in 100% cold methanol (-20 °C) for 5 min. Fixed samples were briefly stained with hematoxylin. This step is required to facilitate the visualization of the worms in an OCT block in future steps. A region of interest was then cut (Supplementary Fig. 3) and placed in a vinyl specimen mold followed by the addition of OCT compound, and immediately frozen. Blocks were stored at -80 °C until tissue sectioning, as is standard for performing ST.

### Total RNA extraction – RNA quality

To verify the RNA quality of the *B. malayi* samples, we extracted RNA from a portion of the worm. Methanol fixed, OCT-embedded *B. malayi* adult female tissue (see section "Parasite material and treatment") was

cryosectioned longitudinally to 8 μm thickness with a CryoStar NX70 cryostat (ThermoFisher). All sections (~11–16 sections per block) throughout 2-3 embedded worm pieces were pooled and collected in Lysing Matrix D tubes (MP Biomedicals, 2 mL, Cat Ref. 6913100). The extraction was done with replicates of the originally embedded blocks and re-embedded blocks where the worm piece was re-embedded onto flattened OCT on a cryo chuck from the same batch as those used for Spatial Transcriptomics. Re-embedding of the sample block was performed to improve the orientation of the block to the sectioning blade to obtain higher quality, intact sections during cryosectioning. The sections in Lysing Matrix D tubes were stored at -80°C overnight. Total RNA was extracted using the RNeasy Plus Mini Kit (Qiagen, Cat No. 74136) with the modifications that 350 μl Buffer RLT Plus + β-mercaptoethanol was added to each Lysing D matrix tube (5 μl of β-mercaptoethanol (2-Mercaptoethanol, Sigma-Aldrich, M6250), added to 500 μl of Buffer RLT Plus (from kit) per sample). The samples were run in a FastPrep-24 5 G (MP Biomedicals) for 40 seconds at 6.0 m/sec, centrifuged for 5 min at 14150 x *g*. The water phase was collected and transferred to a gDNA Eliminator spin column (provided in the kit) that was sealed with parafilm and centrifuged for 30 s at 9800 x *g*. The flow-through containing RNA and proteins was transferred to an eppendorf tube (provided by kit); the beads were added to the same gDNA Eliminator column, and the column sealed with parafilm and centrifuged for 30 s at 9800 x *g*. The flow-through containing RNA and proteins was then transferred to the previous aliquot in the eppendorf tube and 250 μl 96-100% ethanol (Ethanol 96%, VWR, #20823.290 and Ethanol absolute, VWR, #20816.298) was added. The manufacturer's instructions were then followed from the RNeasy Plus Mini Handbook (09/2020) protocol for tissue samples from step 6, including the optional step 10, through step 11 and the RNA was eluted in 10 μl RNase free water (provided by kit). The concentration of extracted total RNA was determined with the RNA HS Qubit assay (Invitrogen by Thermo Fisher Scientific, REF Q32852) following the manufacturer's instructions. Total RNA was diluted, when needed, to between 2-5 ng and RIN values determined using the Agilent RNA 6000 Pico Kit following the manufacturer's instructions. The originally embedded control samples (OE_1 and OE_2) showed an RNA quality of 6-8 RIN and the re-embedded control samples (RE_1 and RE_2) showed an RNA quality of 7-8 RIN, demonstrating that the RNA quality of worms undergoing re-embedding treatment was not impacted by the re-embedding procedure (Supplementary Table 1). Additionally, total RNA from worms under antibiotic doxycycline treatment (~11-21 sections per block) showed an RNA quality of 6-7 RIN in the original embedded sample blocks (D_OE_1 and D_OE_2) and 6 RIN when re-embedded (D_RE_1 and D_RE_2), showing similar RNA quality across embedding strategy and in comparison with control samples (Supplementary Table 1).

### Tissue optimization experiment

To identify the optimal permeabilization time for a specific tissue type for a Visium experiment, a tissue optimization experiment was performed. OCT-embedded *Brugia malayi* adult female whole worms were cryosectioned longitudinally to 10 μm thickness with a CryoStar NX70 cryostat (ThermoFisher) onto Codelink slides containing 100% polyd(T) capture probes. After cryosectioning, all tissue sections were stored at -80°C until experimental processing.

Hematoxylin & Eosin (H&E) was performed as specified in the 10X Genomics Visium Spatial Gene Expression User Guide[39] but skipped step 1.1, Tissue Fixation, since methanol fixation was performed before OCT embedding (see Methods section "Parasite material and treatment"). The slide was removed from the -80°C freezer and placed on dry ice in a sealed container. The slide was warmed at 37°C for 2 minutes on a Thermoblock (ThermMixer with Thermoblock, Eppendorf) with a heated lid (ThermoTop, Eppendorf), then step 1.2.a of the protocol was performed. For step 1.2.j, the slide was incubated in Mayer's Hematoxylin (Agilent Technologies, S330930-2) for 4 minutes at room

temperature. For step 1.2.q, the slide was incubated in Dako Bluing Buffer (Agilent Technologies, CS70230-2) for 1 minute at room temperature. For step 1.2.v, the slide was incubated in Eosin Mix (100 μL Eosin Y Solution + 900 μL Tris-Acetic Acid Buffer (0.45 M, pH 6.0)) for 45 seconds at room temperature. After step 1.2.z, a cover glass (Menzel-Gläser, PN: 12392108, 22x22mm #1, 631-1339) was mounted on the slide with 280 μL of 85% glycerol (Sigma, 49767-100 ML). See Methods section"Microscope & Imaging" for details on the imaging of H&E stained tissue sections.

After imaging, the slide was dipped in RNase and DNase free MQ water in a 0.5 L beaker to remove the cover glass. The slide was air dried, then placed into a prewarmed, 37°C, ArrayIT metallic hybridization cassette (ArrayIt, AHC1X16). Tissue sections were permeabilized by adding 70 μL of pre-warmed (at 37°C) pepsin (Sigma-Aldrich, PN: P7000) in 0.1% in 0.1 M HCl (Honeywell Fluka, PN: 15654960 Fisher Scientific) to each tissue section/slide well and incubating the tissue sections at 37°C on a Thermoblock (ThermMixer with Thermoblock, Eppendorf) with a heated lid (ThermoTop, Eppendorf) for 2 minutes. Pepsin was removed and sections washed with 70 μL SSC 0.1X (diluted with RNase and DNase free MQ water from 20X SSC (Sigma-Aldrich, PN: S66391L)). cDNA reaction mix was prepared with 16 μL 5X FS buffer (Invitrogen, PN: Y02321), 4 μL 0.1 M DTT (Invitrogen, PN: Y00147), 33.3 μL RNase and DNase free MQ water, 8 μL diluted Actinomycin D (10 μL Actinomycin D (Sigma-Aldrich, PN: A1410) in 90 μL RNase and DNase free MQ water), 4 μL NEB BSA (20 mg/mL) (Molecular Biology Grade, New England BioLabs, Catalog # B9000)), dNTP mix (100 μL dATP Solution (100 mM) (Thermo Scientific, Cat No. R0141), 100 μL dGTP (100 mM) (Thermo Scientific, Cat No. R0161), 100 μL dTTP (100 mM) (Thermo Scientific, Cat No. R0171), 10 μL diluted dCTP (5 μL dCTP (100 mM) (Thermo Scientific, Cat No. R0151) in 20 μL RNase and DNase free MQ water), 690 μL RNase and DNase free MQ water), 2 μL diluted Cy3-dCTP (5 μL Cy3-dCTP (PerkinElmer, PN: NEL576001EA) in 20 μL RNase and DNase free MQ water), 4 μL RNaseOUT (Invitrogen, PN: 100000840), and 8 μL SuperScript III (Invitrogen, Cat No. 18080085) per well and pre-warmed at 42°C.

Seventy μL of pre-warmed, 42°C cDNA reaction mix was added and incubated overnight at 42°C on a Thermoblock (ThermMixer with Thermoblock, Eppendorf) with a heated lid (ThermoTop, Eppendorf) for cDNA synthesis. The cDNA mix was removed and each well was washed with 70 μL 0.1X SSC buffer (diluted with RNase and DNase free MQ water from 20X SSC (Sigma-Aldrich, PN: S66391L)). Two drops of RNAScope DAPI (Advanced Cell Diagnostics, PN: 323108) were added to two wells of tissue sections, incubated for 1 minute at room temperature, the DAPI removed, and the wells washed with with 70 μL RNase and DNase free MQ water. The slide was air dried and a cover glass (Menzel-Gläser, PN: 12392108, 22x22mm #1, 631-1339) mounted with 100 μL of SlowFade™ Gold Antifade Mountant (Invitrogen, ThermoFisher Scientific Cat No: S36936). For imaging of DAPI stained tissue sections, a Zeiss AxioImager.Z2 VSlide Microscope using the Metasystems VSlide scanning system with Metafer 5 v3.14.179 and VSlide software was used. The microscope had an upright architecture, and used a widefield system; a 20x air objective with the numerical aperture (NA) 0.80 was used. The camera was a CoolCube 4 m with a Scientific CMOS (complementary metal-oxide-semiconductor) architecture and monochrome with a 3.45 ×3.45 μm pixel size. All DAPI images were taken with a Camera Gain of 6.0 and an Integration Time of 0.02273-0.00072 seconds.

After DAPI imaging, the slide was dipped in RNase and DNase free MQ water in a 0.5 L beaker to remove the cover glass. The slide was air dried, then placed into ArrayIT metallic hybridization cassette (ArrayIt, AHC1X16), and the tissue sections were washed with 70 μL SSC 0.1X (diluted with RNase and DNase free MQ water from 20X SSC (Sigma-Aldrich, PN: S66391L)). Seventy μL of pre-warmed, 56°C Proteinase K (20 mg/mL, Qiagen, Cat No: 19131) in PKD Buffer (Qiagen, Cat No: 1034963) (20 μL Proteinase K in 60 μL PKD Buffer per well) was added

to each well and the slide incubated at 56°C on a Thermoblock (ThermMixer with Thermoblock, Eppendorf) with a heated lid (ThermoTop, Eppendorf) for 1 hour at 300 rpm. Proteinase K was removed and the slide removed from the microarray holder. The slide was placed into a 50 mL falcon tube containing pre-warmed 50°C 2X SSC, 0.1% SDS (diluted with RNase and DNase free MQ water from 20X SSC (Sigma-Aldrich, PN: S66391L) and 10% SDS (Sigma-Aldrich, PN: 71736-100 ML)) for 10 minutes at 50°C and 300 rpm. The slide was then placed into a 50 mL falcon tube containing 0.2X SSC (diluted with RNase and DNase free MQ water from 20X SSC (Sigma-Aldrich, PN: S66391L)) for 1 minute at room temperature and 300 rpm. The slide was transferred to a 50 mL Falcon tube containing 0.1X SSC (diluted with RNase and DNase free MQ water from 20X SSC (Sigma-Aldrich, PN: S66391L)) for 1 minute at room temperature and 300 rpm. The slide was spin dried. An InnoScan 910 Microarray Scanner (Innopsys) and Mapix image analysis software (v. 9.1.0, Innopsys) was used for Cy3 fluorescent slide scanning at 532 nm with an imaging detection gain of 5 and pixel size of 5.0. Example images from the experiment are shown in Supplementary Fig. 4.

We observed that 2 minutes of permeabilization resulted in the capture of mRNA, as seen from the fluorescent Cy3 cDNA footprint, with limited diffusion. We decided to use 2 minutes for our permeabilization time in the Spatial Transcriptomics (Visium) experiments. In addition, we observed that sections with 8 μm thickness improved the section attachment to the slides, and decided to try 8 μm sections with 2 minutes permeabilization in the Spatial Transcriptomics (Visium) experiments.

## Spatial transcriptomics

**Tissue collection.** Methanol fixed, OCT-embedded *Brugia malayi* adult female worm pieces (see section "Parasite material and treatment") were cryosectioned longitudinally to 8 μm thickness with a CryoStar NX70 cryostat (ThermoFisher). Samples BM1 and BM2 were sectioned with the re-embedding technique described in the section "Total RNA extraction - RNA quality" while sample BM3 was sectioned from the original OCT block. All doxycycline treated worms (T1, T2) were re-embedded before sectioning. Re-embedding of the sample block improved the orientation of the block to the sectioning blade and facilitated obtaining higher quality, intact sections during cryosectioning. We collected as many sections as possible (30 total) from the 3 worm control samples (BM1, BM2, and BM3), plus several sections from doxycycline treated worms (T1, T2), onto Visium Spatial Gene Expression Slides (10X Genomics, PN: 2000233). Slides containing tissue sections were stored at -80°C overnight, or up to 72 hours, before experimental processing.

## Modified H&E staining

Spatial Transcriptomics on *B. malayi* adult female reproductive tissue for the control worm samples (BM1, BM2, BM3) was performed as specified by the 10X Genomics Visium Spatial Gene Expression User Guide[39], with the following described modifications. Slides with tissue sections were removed from the -80°C, placed on dry ice in a sealed container, incubated at 37°C for 5 minutes in a Thermoblock (ThermMixer with Thermoblock, Eppendorf) with a heated lid (ThermoTop, Eppendorf), and then placed into an ArrayIT metallic hybridization cassette (ArrayIt, AHC1X16) for steps 1.2.d to 1.2.x of the protocol. Step 1.2.b was prepared as stated. For step 1.2.d, 75 μL of isopropanol (Millipore Sigma, I9516-25ML) was added to each tissue section well, making sure each tissue section was uniformly covered by the solution. Step 1.2.e was performed as stated. Instead of steps 1.2.f-12.g, isopropanol (Millipore Sigma, I9516-25ML) was removed from each well and then each section was washed with 75 μL of RNase and DNase free MQ water, and the wash was repeated two times for a total of three washes. Step 1.2.h was performed as stated. The slide was then warmed at 37°C for 1.5 minutes in a Thermoblock (ThermMixer with

Thermoblock, Eppendorf) with a heated lid (ThermoTop, Eppendorf). For step 1.2.i, 75 μL of Mayer's Hematoxylin (Agilent Technologies, S330930-2) was added to each tissue section well, making sure each tissue section was uniformly covered by the solution. For step 1.2.j, the slide was incubated for 3 minutes at room temperature. Instead of steps 1.2.k-1.2.o, Mayer's Hematoxylin (Agilent Technologies, S330930-2) was removed from each well and then each section was washed with 75 μL of RNase and DNase free MQ water, and the wash was repeated three times for a total of four washes. The slide was then air dried and warmed at 37°C for 1.5 minutes in a Thermoblock (ThermMixer with Thermoblock, Eppendorf) with a heated lid (ThermoTop, Eppendorf). For step 1.2.p, 75 μL of Dako Bluing Buffer (Agilent Technologies, CS70230-2) was added to each tissue section well, making sure each tissue section was uniformly covered by the solution. For step 1.2.q, the slide was incubated for 1 minute at room temperature. Instead of steps 1.2.r-1.2.t, the Dako Bluing Buffer (Agilent Technologies, CS70230-2) was removed from each well and then each well washed with 75 μL of RNase and DNase free MQ water, and the wash was repeated two times for a total of three washes. The slide was air dried and then warmed at 37°C for 1.5 minutes in a Thermoblock (ThermMixer with Thermoblock, Eppendorf) with a heated lid (ThermoTop, Eppendorf). For step 1.2.u, 75 μL of Eosin Mix (100 μL Eosin Y Solution + 900 μL Tris-Acetic Acid Buffer (0.45 M, pH 6.0)) was added to each tissue section well, making sure each tissue section was uniformly covered by the solution. For step 1.2.v, the slide was incubated for 45 seconds at room temperature. Instead of steps 1.2.w-1.2.y, the Eosin mix (100 μL Eosin Y Solution + 900 μL Tris-Acetic Acid Buffer (0.45 M, pH 6.0)) was removed from each well and then each well washed with 75 μL of RNase and DNase free MQ water, the wash was the repeated two times for a total of three washes. The slide was then removed from the ArrayIT metallic hybridization cassette (ArrayIt, AHC1X16) and air dried. Step 1.2.z was performed as stated. A cover glass (Menzel-Gläser, PN: 12392108, 22x22mm #1, 631-1339) was mounted on the slide with 280 μL of 85% glycerol (Sigma, 49767-100 ML).

For antibiotic treated worm T1, the same modified H&E staining was performed as for the control worms, except the Hematoxylin staining was skipped (steps 1.2.i - 1.2.o) and the intermittent warming steps were performed for 3 minutes at 37°C (rather than 1.5 minutes as for the control worms). Slide air drying and intermittent warming steps (the 3 minutes at 37°C steps) were performed with the slide outside of the ArrayIT metallic hybridization cassette (ArrayIt, AHC1X16), with the slide placed back into the cassette before adding the next staining reagent.

For antibiotic treated worm T2, the same modified H&E staining was performed as for the control worms, except the Hematoxylin staining was skipped (steps 1.2.i - 1.2.o).

### Microscope & imaging

For imaging, first an overview 20x image was taken, followed by each individual tissue section imaged with 10 z-stack planes, 2 μm apart at 20x. Hematoxylin & Eosin brightfield images were acquired with a Zeiss Axiolmager.Z2 VSlide Microscope using the Metasystems VSlide scanning system with Metafer 5 v3.14.179 and VSlide software. The microscope had an upright architecture, and used a widefield system; a 20x air objective with the numerical aperture (NA) 0.80 was used. The camera was a CoolCube 4 m with a Scientific CMOS (complementary metal-oxide-semiconductor) architecture and monochrome with a 3.45 ×3.45 μm pixel size. All brightfield images were taken with a Camera Gain of 1.0 and an Integration Time/Exposure time of 0.00004-0.00008 seconds.

### cDNA synthesis & library construction

After imaging, the 10X Genomics Visium Spatial Gene Expression User Guide[39] protocol was resumed from step 2.1.a using a Visium Spatial Gene Expression assay (10X Genomics) kit, with the modifications specified here. For step 2.1.e, the tissue sections were permeabilized for 2 minutes. For step 4.2.d, 19 cycles were used to amplify the cDNA for each control sample subarray, 24 cycles for T1, and 22 cycles for T2. For step 5.5.d, 14 cycles were used for the Sample Index PCR for BM1 and BM2 samples, 15 cycles for sample BM3, 17 cycles for T1, and 18 cycles for T2.

### Sequencing

The average library length was assessed using the BioAnalyzer DNA High Sensitivity kit (Agilent, 5067-4626) on an Agilent 2100 BioAnalyzer. Library concentration was determined with a Qubit dsDNA BR Assay kit (Thermo Fisher Scientific, Q32850). Libraries were diluted to 2 nM, pooled, and sequenced on an Illumina NextSeq 2000 with paired-end, dual index sequencing. Read 1 was sequenced for 28 cycles and Read 2 was sequenced with 150 cycles. Run type and parameters following those specified in the Visium Spatial Gene Expression User Guide sequencing instructions[39].

### Data analysis

**Data pre-processing.** TSO adapter sequences from the 5' end of transcripts and poly(A) sequences from the 3' end of transcripts in the Read 2 raw fastq sequence files were trimmed using cutadapt (v2.3) with a custom bash script (https://github.com/ludvigla/VisiumTrim). TSO sequences were defined as a non-internal 5' adapter with an error tolerance of 0.1, poly-A homopolymers were defined as a regular 3' adapter of 10 As with an error tolerance of 0, and the minimum overlap was defined as 5 bp. Sequence quality was evaluated before and after trimming with FastQC v0.11.8[64] and MultiQC v1.8[65]. 10X Genomics Loupe Browser v4.0.0 was used to manually select spots under tissue sections in the H&E jpeg images. Spots under any portion of tissue were selected to maximize the initial number of spots in the dataset. The resulting json files can be found in the Mendeley dataset specified under "Data Availability".

**Data processing – generation of raw counts.** Genomic sequence and annotation files for *B. malayi* were acquired from WormBase: WBPS14[40] [https://ftp.ebi.ac.uk/pub/databases/wormbase/parasite/releases/WBPS14/species/brugia_malayi/PRJNA10729/] and for *Wolbachia* were acquired as RefSeq: NC_006833.1[66] [https://www.ncbi.nlm.nih.gov/datasets/genome/GCF_000008385.1/]. The *Brugia malayi* genome GFF3 annotation file (brugia_malayi.PRJNA10729.WBPS14.annotations.gff3.gz) was converted to GTF using gffread (with parameters -T -o) from cufflinks (v2.2.1); the GTF file was then filtered to remove all history exons (any exon labeled "history" in the second column of the GTF file). The *Wolbachia* gtf annotation file (GCF_000008385.1_ASM838v1_genomic.gtf.gz) was used to map to *Wolbachia* tRNA and rRNA genes to visualize areas of *Brugia malayi* that contained *Wolbachia*. The *Wolbachia* gtf annotation file (GCF_000008385.1_SM838v1_genomic.gtf.gz) was modified to change "CDS" in column 3 to "exon" to include all genes in the index. 10X Genomics Space Ranger v1.2.0 was used to build a combined *Brugia malayi-Wolbachia* reference with spaceranger mkref and specifying each organism genome fasta file and annotation gtf file as inputs. TSO and poly(A) adapter trimmed paired fastq files were processed with 10X Genomics Space Ranger v1.2.0 with spaceranger count and inputting the corresponding Hematoxylin & Eosin (H&E) images in jpeg format, the custom *Brugia malayi-Wolbachia* reference, and the image manual alignment json file output from 10X Genomics Loupe Browser v4.0.0.

**Quality control – data filtering.** The filtered count matrices (filtered_feature_bc_matrix.h5) and tissue H&E images output from Space Ranger v1.2.0 were analyzed in R (v4.0.3) *Wolbachia* and *B. malayi* genes were separated into different count matrices. *B. malayi* gene types were based on gene annotations (these annotation files can be

found in the github repository specified under "Code Availability"). Protein coding genes were defined as "protein coding" in the "biotype" column of the annotation file. Mitochondrial and ribosomal protein coding genes were defined by the "annotation" column. The *B. malayi* genes were filtered to contain only protein coding genes, and mitochondrial and ribosomal protein coding genes were filtered out. The *B. malayi* data was further filtered by removing spots with fewer than 30 unique genes, fewer than 50 and greater than 10,000 unique transcript molecules (UMIs), more than 3% of UMIs belonging to mitochondrial genes, and more than 30% of UMIs belonging to ribosomal genes and removing genes with less than 1 spot per gene and less than 1 unique transcript molecule (UMI) count per gene. For the post-doxycycline treated worm sample sections, the same filtering criteria were followed as for the control worms, except that spots with fewer than 10 unique genes, fewer than 15 unique transcript molecules (UMIs), more than 5% of UMIs belonging to mitochondrial genes, and more than 35% of UMIs belonging to ribosomal genes were filtered out. The *Wolbachia* count matrix was filtered to remove the same spots as were filtered out from the *B. malayi* count matrix. Data filtering was performed using the *STUtility* package (v1.0)[67] that works on top of the toolkit for single cell analysis Seurat (v3.2.3)[68] in R (v4.0.3).

**Data normalization.** Each worm sample was split into its own filtered count matrix for normalization. Normalization was performed using the SCTransform function from Seurat with default parameters except unique gene counts regressed out and not returning only variable genes (vars.to.regress = "nFeature_RNA", return.only.var.genes = FALSE). After normalization, individual sample count matrices were integrated using Seurat functions SelectIntegrationFeatures with default parameters except "nfeatures = 7000" to acquire the integration features, followed by the PrepSCTIntegration Seurat function with default parameters. MergeSTData was run with default parameters to merge the normalized count matrices into a single object. For the analysis of post-doxycycline treated worms, each tissue section across both treated and control worms was split into its own filtered count matrix for normalization. Normalization was performed with the same parameters as used for the control only worms above, followed by integration of all normalized count matrices into a single object. Normalization was performed using the *STUtility* package (v1.0)[67] that works on top of the toolkit for single cell analysis Seurat (v3.2.3)[68] in R (v4.0.3).

### Clustering analysis
On the SCTransform normalized count matrix, dimensionality reduction was performed using Principal Component Analysis (PCA) with RunPCA on the SCT assay and the integration features (output from SelectIntegrationFeatures function in the "Data Normalization" section) as features. Sample batch effects were removed using RunHarmony[69] (v0.1.0) (group.by.vars and vars_use as worm_sample) applied on the PCA-computed matrix using 10 dimensions, theta of 0, and maximum iteration of 80. Clustering was performed with Seurat functions FindNeighbors, FindClusters, and RunUMAP on the harmony integrated data with 10 dimensions (dims = 1:10), a k parameter of 10 (k.param = 10), and resolution of 0.3 (resolution = 0.3). Clustering analysis was performed using the *STUtility* package (v1.0)[67] that works on top of the toolkit for single cell analysis Seurat (v3.2.3)[68] in R (v4.0.3).

### Differential expression analysis
To find the differentially expressed genes per cluster, the Seurat function "FindAllMarkers" was used with default settings (used default two-sided Wilcoxon Rank Sum test), except for specifying the SCT assay (assay = "SCT") and a log fold change threshold of 0.1 (logfc.threshold = 0.1). Only upregulated and downregulated genes with $p$-value < 0.05 were considered. Differential expression analysis was performed using the *STUtility* package (v1.0)[67] that works on top of the

toolkit for single cell analysis Seurat (v3.2.3)[68] in R (v4.0.3). We used a previous proteomics study[25] (the list of proteins enriched in each dissected tissue can be obtained in the WormMine database) to annotate the different clusters based on the differentially expressed (DE) genes for each cluster. The previous proteomic study indicated a set of proteins that were enriched in specific tissues[25], specifically the body wall, digestive tract, and reproductive tract. The proteins enriched for each of these specific tissue type(s) can be considered as markers for those tissues where they were highly expressed. Therefore, we used these markers to validate our clustering analysis. We looked at the proportion of these tissue specific marker genes in each cluster to assign a cluster to a specific tissue type ((number of tissue specific marker DE genes / total cluster DE genes with adjusted $p$-value < 0.05 and average log fold change > 0) * 100). For *B.malayi* genes, gene annotations and identified orthologs in *C. elegans* were obtained from WormBase[70] [https://wormbase.org/].

### Functional term enrichment analysis
For the functional term enrichment analysis, InterPro description and GO terms for each gene were identified using BioMart[71]. Significantly over-represented functional terms in each cluster were identified using a two-sided Fisher's exact test (FDR < 0.05).

### Co-localization analysis
We considered spots where at least 1 *Wolbachia* rRNA or tRNA gene UMI was detected as *Wolbachia*+ spots and the spots that did not capture *Wolbachia* rRNA or tRNA genes were considered *Wolbachia*- spots. Differential Expression Analysis (DEA) was run between *Wolbachia*+ and *Wolbachia*- spots using the Seurat function "FindMarkers" with both the two-sided Wilcoxon Rank Sum test and two-sided DESeq2 test using default settings, except for specifying the SCT assay (assay = "SCT"), a log fold change threshold of 0.1 (logfc.threshold = 0.1), with ident.1 set to *Wolbachia*+ spots and ident.2 to *Wolbachia*- spots. All DESeq2 DE genes were also found by wilcox, and values from both with adjusted $p$-value < 0.05 are included in Supplementary Data 6. Colocalization analysis was performed using the *STUtility* package (v1.0)[67] that works on top of the toolkit for single cell analysis Seurat (v3.2.3)[68] in R (v4.0.3).

### Soft clustering along *Wolbachia* density & GO enrichment analysis
The MFuzz[72] (v2.50.0) software in R (v4.0.3) was used to perform soft clustering of *B. malayi* genes across *Wolbachia* abundance. *Wolbachia* UMI counts per spot were binned into four groups (none: 0 UMI counts per spot, low: 1-2 UMI counts per spot, medium: 3-9 UMI counts per spot, and high: ≥ 10 UMI counts per spot) and the average expression for each *B. malayi* gene (normalized counts) across all the spots within each *Wolbachia* abundance group was calculated. The resulting dataframe was used as input for MFuzz. Pre-processing was performed by filtering out genes with missing values (filter.NA(eset, thres=0.25) and fill.NA(eset,mode = "knnw")) and filtering out genes with a standard deviation of 0 (filter.std(eset,min.std=0)). Standardization was performed (standardize(eset)), the fuzzifer parameter calculated (mestimate(eset)), and the resulting value (m = 2.54) was used with the Dmin function to get an estimate of an optimal number of soft clusters. The optimal number of clusters, 4, was further determined based on the Gene Ontology (GO) terms output from topGO[73] (v2.42.0). MFuzz soft clustering was performed (mfuzz(eset, c = 4, m = 2.54)) and genes with memberships scores ≥ 0.7 were considered core cluster genes (Supplementary Data 6A-D). Soft clusters were termed "Patterns" to avoid confusion with the ST clusters. All genes used as input for MFuzz were run through WormBase ParaSite[74] (WBPS18) [https://parasite.wormbase.org/] using biomaRt[71] (v2.46.3) in R (v4.0.3) to acquire the GO terms associate with each gene. All genes used as input for MFuzz were considered background genes for running GO term enrichment

analysis. GO term enrichment analysis of the biological porocesses (ontology = "BP") for the core genes for each cluster (Pattern) was run and unadjusted p-values for the top GO terms were determine from a two-sided Fisher's exact tests in topGO[73] (v2.42.0) in R (v4.0.3) (Supplementary Fig. 2, Supplementary Data 7A-D). We left p-values unadjusted as topGO documentation mentions adjusted p-values may be misleading.

### Post-doxycycline treated worm analysis

The fold change differences between treated versus control worms and *Wolbachia*+ versus *Wolbachia*- spots within treated worms across 65 genes (the 65 differentially expressed genes between *Wolbachia*+ and *Wolbachia*- spots identified by the co-localization analysis of the control worms) were determined using Seurat function FoldChange with default parameters. Post-doxycycline treated worm analysis was performed using the *STUtility* package (v1.0)[67] that works on top of the toolkit for single cell analysis Seurat (v4.1.0)[68] in R (v4.0.5).

### 3D figure

The tissue images used in the alignment were pre-processed using Adobe Photoshop 2022 (v23.2.1) to remove background and to separate each section of the worm into one image. The posterior and anterior regions of the separated worm sections were manually aligned using 'ManualAlignImages()' and the 3D model was made using 'Create3DStack()' from *STUtility* package (v1.0). For worm BM2, sections 1-9 were aligned to section 10 for generating the 3D model. Section 11 was not used in the 3D model for BM2 due to the coordinates for this section being significantly distant (outside the allowed limit of rotation and shifting coordinates) to be aligned over the center section in the 3D alignment. The 3D images for the genes of interest were generated using 'FeaturePlot3D()', also from the *STUtility* package (v1.0) in R (v4.2.0).

### Shiny app

The gene expression heatmap images on the shinyapp were generated using "ST.FeaturePlot()" (dark.theme = T for better color contrast to aid the visualization) and cluster images using "FeatureOverlay()" from the *STUtility* package (v1.0) in R (v4.2.0). Shiny theme 'sandstone' was applied as part of the overall design aesthetics. The app is hosted on https://www.shinyapps.io for public access.

### Statistics and reproducibility

The spatial transcriptomics data presented in the Main and Supplementary Figures are generated from n = 3 biological replicates of *Brugia malayi* adult female control worms and n = 2 biological replicates of *B. malayi* adult female post-doxycycline treated worms. Each n refers to a distinct *B. malayi* adult female worm sample, and from each sample multiple tissue sections were collected for Spatial Transcriptomics (Visium Spatial Gene Expression assay, 10X Genomics).

### Reporting summary

Further information on research design is available in the Nature Portfolio Reporting Summary linked to this article.

## Data availability

The raw sample sequence fastq files generated in this study have been deposited in the NCBI SRA database under accession code PRJNA870734[75]. The processed gene count matrices, related metadata, corresponding ST tissue H&E microscopy images, and 3D model HTML files generated in this study have been deposited in the in Mendeley Data[76] (https://doi.org/10.17632/8f62vydg3z.1). The gene expression information for all the *B. malayi* genes from the normalized SCT assay are available for visualization on our publicly available app: https://giacomellolabst.shinyapps.io/brugiast-shiny/. *Brugia malayi* genome assembly PRJNA10729 WormBase release WBPS14[40] was used.

*Wolbachia* genome assembly was acquired as RefSeq: NC_006833.1[66] (https://www.ncbi.nlm.nih.gov/datasets/genome/GCF_000008385.1/). *B.malayi* gene annotations and identified orthologs in *C. elegans* were obtained from WormBase[70] (https://wormbase.org/). WormBase ParaSite[74] (WBPS18) (https://parasite.wormbase.org/) was used to get Gene Ontology (GO) Terms.

## Code availability

The scripts used to generate count matrices from raw sequence fastq files and related R scripts used to analyze count matrices for quality control and filtering, normalization, clustering and differential expression analysis, colocalization analysis, 3D figure generation, and treated worm analysis can be accessed from our Github repository[77] (https://github.com/giacomellolab/Brugia_malayi_study) and from Zenodo[78] (https://zenodo.org/record/8347641).

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

## Acknowledgements

This work was supported in part by the Division of Intramural Research (DIR) of the NIAID/NIH (D.V., E.G.). S.G. was supported by the Swedish Research Council VR grant 2020-04864. The *Brugia malayi* parasites (NR-48892) were provided by the NIH/NIAID Filariasis Research Reagent Resource Center for distribution through BEI Resources, NIAID, NIH. The computations and data handling of the *Brugia malayi* sequencing data were enabled by resources provided by the Swedish National Infrastructure for Computing (SNIC) at UPPMAX, partially funded by the Swedish Research Council through grant agreement no. 2018-05973. We thank Ludvig Larsson for his discussions on analysis of the Spatial Transcriptomics data.

## Author contributions

H.S. and D.V. performed investigatory experiments, developed methodology, analyzed and visualized the data, worked on validation, curated the data, conducted project administration, and wrote the manuscript. H.S. and M.C. implemented software for formal analysis of the data. Y.M. generated the 3D visualization, set up the shiny app analyses, and gave input for the formal analysis of the data. S.S. aided in methodology and software implementation for formal analysis of the data. S.G. and E.G. conceptualized, designed, led and supervised the study, developed methodology, acquired funding and resources, conducted project administration, and wrote and edited the manuscript.

## Funding

## Competing interests

H.S., Y.M., S.S., S.G. are scientific advisors to 10X Genomics, Inc. that holds IP rights to the ST technology. S.G. holds 10X Genomics stocks. The remaining authors declare no competing interests.
