## [Peer Review File · Nature Communications]

Miniature spatial transcriptomics for studying
parasite-endosymbiont relationships at the micro scaleREVIEWER COMMENTS

Reviewer #1 (Remarks to the Author):

The present study has focused on studying a posterior region of human filarial nematode *Brugia malayi* that contains the reproductive tissue and developing embryos. To spatially explore how *Wolbachia* interacts with the worm's reproductive system, the authors carried out spatial characterization using Spatial Transcriptomics (ST) across this region of interest and have provided a proof-of-concept for miniature-ST to explore spatial gene expression patterns demonstrating the usefulness of this technique. The authors state that this approach may open up new scenarios for studying infectious diseases caused by filarial or other similar worms.

Reviewer's comments:

The study is essentially a technical advance as the authors have made several technical changes and thus overcome the challenges in the Visium protocol to obtain spatially resolved transcriptomic information from *Brugia malayi*. Though concise, and straight forward, the cost, expertise and set-up needed to carry out such a task will be a limiting factor for the wide applicability of this technique. This and some other drawbacks of the adopted approach limit the publication of this piece of work in the present form. The authors need to address the following concerns.

Major concerns:

1. The study is only exploratory in nature. It describes the application of Spatial Transcriptomics to explore spatial gene expression patterns across different clusters in *B. malayi*, but does not essentially provide any novel experimental data that can be put to beneficial use in terms of developing therapeutics for this neglected disease. The authors have only stated purported functions of various genes identified using ST, but have not described the actual role of the different genes identified through this technique in Brugian biology. Most of the functions described are already reported in the literature. There are no mechanistic details provided that actually demonstrate how brugian and wolbachial genes work in unison to support each other survival. Thus, barring a technical advance, the manuscript does not provide any new or novel information.
2. The study has essentially focused on the adult worm. It would have been good if other life stages of the adult worm (*Microfilariae*-Mf and infective larvae-L3) would have been taken into account. This would have provided information about the changes in the gene expression profile during the molting process, and essentially identified key "conserved" genes that could be targeted for therapeutic benefits.
3. Similarly, as in point 2 above, it would be good to have ST data at hand collected before and after the chemotherapeutic treatment of the worm.

Minor concerns:

1. Parasites were recovered from infected Gerbils. How old was the infection in Gerbils? This might affect the development stage/sexual maturity of the worm even though the authors mention that they took worms that were on average 150 μm in diameter and 43-55 mm in length.
2. The sample size for re-embedding was two worms, while only one worm was not re-embedded prior to sectioning. This small sample size may be too small, the authors may discuss this point.

Reviewer #2 (Remarks to the Author):

Peer review report NCOMMS-22-42920

In this work, Sounart et al reported the development of an experimental approach to study parasite-endosymbiont interactions using miniature spatial transcriptomics. The authors report novel findings regarding the spatial distribution of tissue-specific genes and report the co-localisation of the endosymbiont *Wolbachia* within discrete tissue areas. I would like to highlight the importance of this study and the major implications that such approaches will have in our understanding of filarial biology. The manuscript is presented concisely, but at times it is unclear if some of the statements are substantiated by experimental evidence or are hypotheses yet to be experimentally tested. Similarly, there are multiple instances where references to previous work are needed, for clarity. Overall, I found this to be an exciting manuscript and would like to congratulate the authors for this work. I hope my comments help improve the manuscript.

Abstract and introduction

Well balanced introduction to the topic. Just some minor points

Line 70: consider an alternative word than "cultured". Maybe maintained? Passaged?

Line 112: To my knowledge, there are at least three additional studies reporting the use of spatial approaches to study infections, including tuberculosis, *Trypanosoma brucei*, and *Plasmodium berghei*. Consider citing these articles to capture the breadth and impact of such approaches to study parasitic infection.

Results

1. Given the importance of visualising the tissue prior to library preparation, and the novelty of the method presented here, I wonder if the authors could include H&E staining of the sections used for ST? Alternatively, a representative image would suffice.
2. Did the authors check RNA integrity post-fixation? I appreciate this might be difficult, but wondered if this could be included as supplementary, to increase the confidence in this new method.
3. I wonder whether the authors conducted a permeabilisation assay to determine the optimal time for capturing polyadenylated transcripts? If so, I believe this should be included as a supplementary file.
4. In line 160, the authors mention morphological annotations, but I failed to find such information. Could this be included in the figure? I believe this is an important piece of information.
5. In figure 1C, can the authors clarify what the different numbers per sample refer to? Are these sections? If so, why there are some sections missing? The numbers do not seem consecutive (e.g., BM2_1 is missing).
6. For figure 2B, can the authors generate a spatial feature plot for genes of interest identifying each of the tissue sections reported in 2C? I believe this will significantly strengthen the point made regarding the spatial identification of various tissues within the worm.
7. Some of the spots do not seem to be covered by tissue in figure 2B, which might be due to over-permeabilisation. Can the authors comment on this?
8. Although exciting, it is unclear to me how the data reported in figure 2D was generated. I suggest that all the sections captured prior to generation of the 3D model should be provided as a separate supplementary file to make it easier to understand. This could perhaps be added to supplementary information 1.
9. Given the importance of the conclusions presented in figure 2, the authors should validate some of their markers using alternative approaches. For instance, an in situ hybridisation approach could certainly help to demonstrate that the novel markers identified with the ST approach are indeed tissue-specific. This 2D approach would complement efforts to generate a 3D atlas.
10. I am having some difficulties with figure 2E. Most of the pathways shown here are associated with ribosomal metabolism and ubiquitin, and seem to show little cluster specificity, especially for clusters 1 and 2. The pathways reported in figure 2E do not necessarily align with those mentioned in the text (e.g., I could not find sterol-binding and transfer activity in 2E). Similarly, I could not see the gene pathway associated with ShKT-domain containing proteins in the figure. Although I

appreciate these are reported in the supplementary tables, perhaps including these in figure 2E would make it easier for a reader to interpret. I believe the data presented in supplementary information 1 should be moved to the main figure as it makes the point clearer. However, additional experiments validating some of these markers, either via FISH or IFA (if not reported elsewhere), should be included in the manuscript.

11. In figure 3, can the authors co-detect/correlate the expression of the selected glycolytic genes with the presence of Wolbachia? Would an imaging experiment help to substantiate the claims made in this figure?

12. In line 405 - 409: Is this statement based on a prediction from the data, or on previously reported work? I cannot seem to follow the logic of these statements. Can the authors clarify?

13. Although I appreciate the observations and data interpretation in figure 4C and D are exciting, I believe it is a bit speculative and would suggest that the authors rephrase this section to state that this is something that needs to be experimentally validated and remains hypothetical.

14. I think it would be critical to specifically highlight which novel genes have been discovered with this approach and take them forward for validation. I appreciate this atlas will be a great resource to the community, but it is imperative to provide some kind of additional validation to support the claims, especially around genes/pathways that might have downstream implications for parasite control (e.g., interaction with Wolbachia).

15. In addition to the miniature ST approach, the 3D model is quite interesting, but I feel like it has not been put forward as much as it should have in the main text and figures. I would suggest that the authors consider this point and edit the figures to incorporate some of the 3D data, as in my opinion this is one of the strongest points of the manuscript.

Minor points:

- Some references are missing to support the statements (e.g., line 281-282, and again in 285 - 286, just to cite a few). Please check throughout.

Discussion

Minor points:

- Can the authors comment on how their work compares to, or enrich previously published work reporting the use of spatial approaches to characterise antiparasitic targets in *B. malayi*?

Reviewer #3 (Remarks to the Author):

The study presented by Sounart et al is intriguing and potentially quite novel and impactful. However, I think it requires some significant clarification of various aspects and major revisions. I have structured my review into the following categories.

1. Significance and noteworthy results:

This study proposes technical and biological advances. On the technical side, the authors present a miniaturised-spatial transcriptomics approach for use with small tissue samples. However, I don't think this advance's impact is sufficiently explained. The authors mention that their samples are relatively small, but they work with 150-200µm tissue samples from worm sections that are several millimetres long. Numerous studies have applied spatial transcriptomics to study similarly sized or smaller tissues, including mouse or *Drosophila* embryos or *C. elegans* tissues.

See, for example:

Rödelsperger, Christian, et al. "Spatial transcriptomics of nematodes identifies sperm cells as a source of genomic novelty and rapid evolution." *Molecular biology and evolution* 38.1 (2021): 229-243.

Peng, Guangdun, et al. "Spatial transcriptome for the molecular annotation of lineage fates and cell identity in mid-gastrula mouse embryo." *Developmental cell* 36.6 (2016): 681-697.

Srivatsan, Sanjay R., et al. "Embryo-scale, single-cell spatial transcriptomics." *Science* 373.6550 (2021): 111-117.

The authors also note that head tissue spatial transcriptome has already been published for *Brugia malayi*. However, they do not explain why these method could not be applied to their study.

Biologically, *Brugia malayi* is a major global parasite and understanding its interactions with *Wolbachia* is of significant interest. However, I think the authors need to specify the significance of their study more clearly. The authors allude to this significance but they do not clearly explain it, in my opinion. For example, lines 98-100 state that 'Resolution of *Wolbachia* and *B. malayi* gene expression at the level of individual cells or tissue morphological regions is needed to design more effective therapeutic strategies for lymphatic filariasis. This may be the case, but it is not self-evident how and needs to be expanded upon. Likewise, it is not clear why and in what ways the authors' spatial transcriptomic data provide an advance over existing tissue-specific or dual transcriptomics studies exploring *Brugia* and *Wolbachia*. Again, this isn't to say that the study isn't providing valuable new data, but rather, the need to clarify how the study provides a major biological advance, separate from the method.

2. Originality: To my knowledge, this is the first study to apply spatial transcriptomics to explore *Brugia* tissues (beyond the head tissue study the authors cited) or the interactions between *Wolbachia* and its host. I am unaware of anyone using the specific method the authors describe for their spatial transcriptomics experiments. However, as above, I think the authors need to explain more thoroughly what their method makes possible that existing spatial transcriptomics methods do not. In addition, I agree that the spatial data are novel, but how novel? What is now understood about this interaction that could not have been gleaned from existing studies and what is the impact of this new understanding?

3. Are the conclusions supported: The technical conclusion centres around the assertion that the method produces high-quality spatial transcriptomic data from small samples. This statement is somewhat subjective. For example, how does one define a 'small' sample? I agree that the method appears to work well and suitably characterises the tissues and samples the authors have included in their study. However, whether the method is high impact depends on whether it's a major advance over existing methods. The samples the authors have assessed do not appear to be notably smaller than those analysed using existing methods for other organisms, and the authors don't explain why the method already used for *Brugia* head tissues is unsuitable. More detail on this would be very helpful.

Capturing *Wolbachia* transcripts that are retained despite the polyA-based enrichment of the *Brugia* mRNAs is a clever approach. Coupling this with spatial transcriptomics allows the authors to correlate localised changes in the *Brugia* transcriptome and the presence or absence of the symbiont. I quite like this and think it is a novel and exciting aspect of the paper that will be of broad interest.

Characterising the *Wolbachia* transcriptome: the authors allude to being able to use their approach to undertake spatial resolution of *Brugia* and *Wolbachia*. On the latter, I think this is less clear and may require additional analysis. The authors captured *Wolbachia* mRNAs as by-catch. However, the assertion that they 'spatially resolve' the *Wolbachia* transcriptome is a bit ambiguous. If the authors mean that they were able to identify the location of *Wolbachia* and see how this correlated with differences in the host transcription, I agree. However, if they mean they could resolve the *Wolbachia* transcriptome at a spatial level, I think the current analyses can't conclude this. Maybe this is not what the authors meant. If it is, I think they need to provide more evidence. For example, they could compare the by-catch transcriptome with existing dual-RNAseq studies to see if the *Wolbachia* transcriptome is being replicated or look at transcript coverage metrics and other parameters commonly applied to quality control conventional RNAseq datasets.

4. Study design and methodology: As noted above, the study design and methodology appear sound and well-controlled. The analyses are appropriate and, except for my query regarding resolving the *Wolbachia* transcriptome (and my apologies if I have misunderstood here), support

the study's conclusions. The methods sections are well described, and custom scripts, raw data and the various outputs of the experiments are provided and sufficiently well described in the methods, supplementaries, and through the linked URLs the authors have provided.

Overall, the approach is interesting, and the potential to apply spatial transcriptomics to study host-endosymbiont interactions is powerful and exciting. I think the study is well-executed and sound and may well be high-impact. However, this impact needs to be more thoroughly explained. The authors need to explain what their method does that current approaches cannot. Similarly, what are the overall implications of the new transcriptomic data for *Brugia* or *Wolbachia* that it provides, and does this provide major insight into the fundamental biology of the interaction or new avenues to control the parasite? Based on this, I'd prefer the article to be significantly revised but still considered for publication. I'd be happy to review a revised submission if that is required.

POINT-BY-POINT REPLIES TO REVIEWERS' COMMENTS

Reviewer #1 (Remarks to the Author):

*The present study has focused on studying a posterior region of human filarial nematode *Brugia malayi* that contains the reproductive tissue and developing embryos. To spatially explore how *Wolbachia* interacts with the worm's reproductive system, the authors carried out spatial characterization using Spatial Transcriptomics (ST) across this region of interest and have provided a proof-of-concept for miniature-ST to explore spatial gene expression patterns demonstrating the usefulness of this technique. The authors state that this approach may open up new scenarios for studying infectious diseases caused by filarial or other similar worms.*

Reviewer's comments:

*The study is essentially a technical advance as the authors have made several technical changes and thus overcome the challenges in the Visium protocol to obtain spatially resolved transcriptomic information from *Brugia malayi*. Though concise, and straight forward, the cost, expertise and set-up needed to carry out such a task will be a limiting factor for the wide applicability of this technique. This and some other drawbacks of the adopted approach limit the publication of this piece of work in the present form. The authors need to address the following concerns.*

Thank you for your comments on the work, such as its concise and straightforward implementation for spatially resolving *Brugia malayi*. Given that the approach implements and modifies the commercially available 10X Genomics Visium platform, and that we were able to reproduce our results through the steps detailed in the methods, we do not envision that the approach would be limited by the need for extensive expertise and set-up. We hope the cost of commercially available spatially-resolved transcriptomics methods, such as Visium, will lower in the future. In addition, our miniature-ST approach of constructing a tissue atlas from small-scaled tissue is also economically advantageous, since all tissue sections can be collected for one sample on a single Visium capture area. We have addressed the additional concerns mentioned by the Reviewer in our responses below.

Major concerns:

*1. The study is only exploratory in nature. It describes the application of Spatial Transcriptomics to explore spatial gene expression patterns across different clusters in *B. malayi*, but does not essentially provide any novel experimental data that can be put to beneficial use in terms of developing therapeutics for this neglected disease. The authors have only stated purported functions of various genes identified using ST, but have not described the actual role of the different genes identified through this technique in Brugian biology. Most of the functions described are already reported in the literature. There are no mechanistic details provided that actually demonstrate how brugian and wolbachial genes work in unison to support each other survival. Thus, barring a technical advance, the manuscript does not provide any new or novel information.*

We would like to emphasize that the novelty presented here is the spatial localization of the whole transcriptome within the worm tissue region of interest and that we can spatially map different

regions within a small worm. Previous techniques were able to perform a spatial reconstruction along the anterior-posterior (A/P axis) in several small organisms, as discussed in the Introduction in lines 84-93, but only by homogenizing the transcriptome profiles across cell types within each transverse section. These previous approaches often worked with smaller organisms, such as mouse (Srivatsan et al. 2021; Peng et al. 2020) and *Drosophila* embryos (Combs and Eisen 2013), with similar length-width ratios that can be $> 500 \mu\text{m} \times 500 \mu\text{m}$ (depending on the developmental stage) (Srivatsan et al. 2021). Filarial worms pose a unique challenge with very different length-width ratios ($< 150 \mu\text{m}$ wide and millimeters in length) and long, multinucleated hypodermal cord cells that make single-cell RNAseq unfeasible. In our work, instead, we focus on a specific region of the worm and spatially resolve different structures within this region in a data-driven manner. To our knowledge, this is the first time this is done in small, disease-causing worms utilizing a commercially available platform where intact tissue sections can be explored. Thus, generating this resource of a specific region of interest in *Brugia malayi* adult female worms is in itself novel, and the community can further explore the data and probe different aspects more deeply in future work to ask specific biological questions. To this end, we provide a valuable resource for the Brugian scientific community that can be explored by using our Shiny app (<https://giacomellolabst.shinyapps.io/brugiast-shiny/>).

In terms of mechanistic details on how *B. malayi* and *Wolbachia* genes work in unison, we co-localize the expression of *B. malayi* genes in tissue areas with and without *Wolbachia*. Moreover, we compare the expression of parasite genes across different tissues that contain different levels of *Wolbachia*. For example, in the Results section “**Spatial localization of key *B. malayi* glycolytic enzyme genes**”, lines 339-375, we analyze the expression of parasite glycolytic enzymes in different tissues, and discuss how such genes are associated with *Wolbachia* presence in these tissues in lines 397-406. Conducting additional conditional/functional experiments to look into specific genes and/or processes and their mechanistic details are to be done in follow-up studies.

To address the Reviewer’s comment on enriching the biological component of our study, we have now added a comparison to antibiotic treated worms in lines 431-469, which shows that worms under doxycycline treatment had fewer *Wolbachia* and expression shifts in genes identified in the control worms to co-localize with *Wolbachia*. Lower *Wolbachia* levels after antibiotic treatment are expected based on previous studies (as mentioned in the manuscript). Almost all of the 65 differentially expressed genes between *Wolbachia*+ and *Wolbachia*- spots identified previously in the control worms showed lower average expression in the treated worms compared to control worms, including the downregulation of heme-associated genes that are potentially involved in the mutualism between *Wolbachia* and *Brugia* parasites (Luck et al. 2016). We also observed the upregulation of certain protease genes that are likely involved in the degradation of dying *Wolbachia* in *Wolbachia*+ spots compared to *Wolbachia*- spots of treated samples. These results provide a preliminary localization of gene-specific, doxycycline-induced effects on the *B. malayi* - *Wolbachia* relationship.

Overall, our work is a significant contribution where we demonstrate a unique approach of untargeted, whole-transcriptome capture in intact, longitudinal tissue sections of an under-studied *B. malayi* region. This is combined with co-localization of the bacterial endosymbiont, *Wolbachia*, at $55 \mu\text{m}$ spatial resolution that distinguishes different tissue structures and transcripts, and

facilitates a 3D reconstruction of the region of the worm where transcripts can be explored along multiple axes (anterior-posterior, ventral-dorsal). Finally, our miniature-ST approach enables a novel method to probe very small organisms, such as parasitic worms, rather than large tissues, opening up the possibility for a whole array of new studies in small worms.

2. The study has essentially focused on the adult worm. It would have been good if other life stages of the adult worm (Microfilariae-Mf and infective larvae-L3) would have been taken into account. This would have provided information about the changes in the gene expression profile during the molting process, and essentially identified key “conserved” genes that could be targeted for therapeutic benefits.

We thank the Reviewer for the suggestion and we agree that it would have been very interesting. However, there is an important point to consider for implementing our approach on microfilariae and larvae stages of the worm. The goal of our spatially resolved approach is to distinguish spatial gene expression patterns in morphologically distinct tissues. In microfilariae and larvae (L3), there are not many morphologically distinguished tissues, for example, there are no gut or uterus tissues. Thus, there would not be many spatially distinct structures to resolve. Consequently, bulk RNA-sequencing makes the most sense, both technically and cost-wise, for that stage of the organism. Microfilariae and larvae are even smaller (length = 175 - 230 μm , width = 5 - 7 μm) than the adult worms and the technique we developed already pushes the limit of resolution (55 μm). Performing Visium at the current resolution of 55 μm would result in very few spots covering Mf, which will not give any resolution improvements compared to bulk RNA-seq of Mf. Therefore, attempting to use the miniature-ST technique for even smaller organisms is not yet possible.

3. Similarly, as in point 2 above, it would be good to have ST data at hand collected before and after the chemotherapeutic treatment of the worm.

We thank the Reviewer for bringing up this point and agree that studying the spatial gene expression patterns of the worm after chemotherapeutic treatment would be a valuable contribution. To address the Reviewer’s comment, we have added to the manuscript the section “**Spatial transcriptomic patterns in post-doxycycline treated worms**” in lines 431-469 with the accompanying **Figure 6** to describe some preliminary spatial data from *B. malayi* worms after treatment with doxycycline (12.5 μM , in vitro 3 days). In summary, and as described in response to comment #1, we observed that worms under doxycycline treatment had lower levels of *Wolbachia* and expression shifts in genes identified in the control worms to co-localize with *Wolbachia*. For example, treated worms showed lower expression of heme-associated genes that are potentially involved in the mutualism between *Wolbachia* and *Brugia* parasites (Luck et al. 2016) and an upregulation of certain protease genes that are likely involved in the degradation of dying *Wolbachia* in treated samples *Wolbachia*+ spots compared to *Wolbachia*- spots. Such results show the spatial localization of gene-specific, doxycycline-induced effects on the *B. malayi* - *Wolbachia* relationship.

Minor concerns:

1. Parasites were recovered from infected Gerbils. How old was the infection in Gerbils? This might affect the development stage/sexual maturity of the worm even though the authors mention that they took worms that were on average 150 μ m in diameter and 43-55 mm in length.

B. malayi is required to grow in a mammalian host to reach the adult stage. Gerbils are the standard organisms that these worms are grown in laboratory conditions. In this study, we used mature adult female worms (>120 dpi), as stated in the manuscript Methods section in line 618. Mature females have fully developed gonads, are fertile, and already produce microfilariae (offspring). A study on *B. malayi* worms grown in *Meriones unguiculatus* gerbils—using the same FR3 strain in the peritoneal cavity, from the same facility where the worms used in our study were acquired—found that at 71 dpi, all adult female worms were fertilized and had microfilariae present in the uterus.

2. The sample size for re-embedding was two worms, while only one worm was not re-embedded prior to sectioning. This small sample size may be too small, the authors may discuss this point.

Thank you for bringing up the important point of sample size regarding the embedding of the worms for cryosectioning. We tested if the quality of the RNA was impacted by the re-embedding procedure by collecting total RNA from pools of 2-3 samples per original embedded (i.e. not re-embedded) and re-embedded worms in duplicate and found the RNA quality (RIN values) to be similar with 6-8 RIN values for the originally embedded worms and 7-8 RIN values for the re-embedded worms. Please see the methods section “**Total RNA extraction - RNA quality**” in lines 628-665 and the added **Supplementary Table 1** (also below). In addition, in the Visium experiments, even though the sample size is smaller, two different re-embedded worm samples were used as biological replicates and showed consistent results between one another and to the originally embedded worm sample (as seen in **Figure 1B-D**), thus the re-embedding of the worm does not seem to impact the gene expression information captured by the method. The purpose of the re-embedding was to improve the orientation of the OCT embedded sample block in regard to the cryosectioning blade for obtaining intact and flat sections and thus improve the quality of the sections obtained for the method. Since the re-embedding was for the reorientation of the sample to collect higher quality sections, we did not separate the worms by this factor in the downstream analysis. We added an additional explanation of this point in the main text results section “**Miniature-ST provides reproducible capture of gene expression information**” in lines 156-160, and in the methods section under “**Total RNA extraction - RNA quality**” in lines 628-665, and under “**Spatial Transcriptomics**” and “**Tissue collection**” in lines 753-758. We also added this information to the Discussion in lines 490-497.

Sample name	RIN	Description
OE_1	8	Total RNA from 2 worms from originally embedded blocks
OE_2	6	Total RNA from 3 worms from originally embedded blocks
RE_1	7	Total RNA from 2 worms from re-embedded blocks

RE_2	8	Total RNA from 3 worms from re-embedded blocks
D_OE_1	7	Total RNA from 2 doxycycline treated worms from originally embedded blocks
D_OE_2	6	Total RNA from 2 doxycycline treated worms from originally embedded blocks
D_RE_1	6	Total RNA from 2 doxycycline treated worms from re-embedded blocks
D_RE_2	6	Total RNA from 2 doxycycline treated worms from re-embedded blocks

Supplementary Table 1. RNA quality of total RNA from original and re-embedded worms from control and treated worms. Quality of total RNA, measured as RIN value, extracted from worms in the original OCT embedded blocks for control worms (OE_1 and OE_2) and doxycycline treated worms (D_OE_1 and D_OE_2), and from the re-embedded blocks for control worms (RE_1 and RE_2) and doxycycline treated worms (D_RE_1 and D_RE_2).

Reviewer #2 (Remarks to the Author):

Peer review report NCOMMS-22-42920

In this work, Sounart et al reported the development of an experimental approach to study parasite-endosymbiont interactions using miniature spatial transcriptomics. The authors report novel findings regarding the spatial distribution of tissue-specific genes and report the co-localisation of the endosymbiont Wolbachia within discrete tissue areas. I would like to highlight the importance of this study and the major implications that such approaches will have in our understanding of filarial biology. The manuscript is presented concisely, but at times it is unclear if some of the statements are substantiated by experimental evidence or are hypotheses yet to be experimentally tested. Similarly, there are multiple instances where references to previous work are needed, for clarity. Overall, I found this to be an exciting manuscript and would like to congratulate the authors for this work. I hope my comments help improve the manuscript.

Thank you for your highlights of the important contributions of this study, such as its implications in applying our approach to understanding filarial biology. We are happy to hear the Reviewer is excited by the potential of this work. We have responded to the Reviewer's concerns on the lack of clarity of our statements being supported by experimental evidence versus hypotheses that remain to be tested, and included additional references for improved clarity, in the comments that follow.

Abstract and introduction

Well balanced introduction to the topic. Just some minor points

Line 70: consider an alternative word than "cultured". Maybe maintained? Passaged?

Thank you for the feedback on the word choice. We have changed the text from "as it can be cultured across" to "as it can develop from later larval (L3) to adult life cycle stages" on line 46.

Line 112: To my knowledge, there are at least three additional studies reporting the use of spatial approaches to study infections, including tuberculosis, Trypanosoma brucei, and Plasmodium berghei. Consider citing these articles to capture the breadth and impact of such approaches to study parasitic infection.

Thank you for your input on including these studies in the manuscript, we agree that they emphasize the impact spatial approaches have on studying parasitic infections. We have added the articles to the Introduction section of the manuscript and briefly discuss the contributions of these studies in lines 67-72.

Results

1. Given the importance of visualising the tissue prior to library preparation, and the novelty of the method presented here, I wonder if the authors could include H&E staining of the sections used for ST? Alternatively, a representative image would suffice.

We agree with the Reviewer on the importance of visualizing the tissue morphology to link histology with spatial gene expression patterns. We have now added a new **Supplementary Figure 1** (also below) that contains the H&E images of the tissue sections of the three analyzed worms and two images at larger magnification with different tissue structures annotated (**Supplementary Figure 1D**). Moreover, the full H&E images are provided in the Mendeley dataset under Reserved DOI: [10.17632/8f62vydg3z.1](https://doi.org/10.17632/8f62vydg3z.1) that will become publicly available.

Supplementary Figure 1. H&E images and morphology of study tissue sections. H&E images of the tissue consecutive sections from worm samples BM1 (A), BM2 (B), and BM3 (C) used in the study. (D) Zoom-in of tissue section BM1_3 and BM1_7 accompanied by morphological annotations. Magnification 20x.

2. Did the authors check RNA integrity post-fixation? I appreciate this might be difficult, but wondered if this could be included as supplementary, to increase the confidence in this new method.

We agree with the Reviewer that it is important to check the RNA integrity after fixation. As described in our Methods section under “**Parasite material and treatment**” in lines 615-626, the worms are fixed in methanol before being embedded in OCT. Therefore, the sections used for the spatial transcriptomic (Visium) experiments and used for measuring RNA integrity were from post-fixation worms as described in the Methods section under “**Total RNA extraction - RNA quality**” in lines 628-665. However, to make this information more readily accessible to the reader, we have added a new **Supplementary Table 1** (also below) to show the RIN values for worms processed in this manner.

Sample name	RIN	Description
OE_1	8	Total RNA from 2 worms from originally embedded blocks
OE_2	6	Total RNA from 3 worms from originally embedded blocks
RE_1	7	Total RNA from 2 worms from re-embedded blocks
RE_2	8	Total RNA from 3 worms from re-embedded blocks
D_OE_1	7	Total RNA from 2 doxycycline treated worms from originally embedded blocks
D_OE_2	6	Total RNA from 2 doxycycline treated worms from originally embedded blocks
D_RE_1	6	Total RNA from 2 doxycycline treated worms from re-embedded blocks
D_RE_2	6	Total RNA from 2 doxycycline treated worms from re-embedded blocks

Supplementary Table 1. RNA quality of total RNA from original and re-embedded worms from control and treated worms. Quality of total RNA, measured as RIN value, extracted from worms in the original OCT embedded blocks for control worms (OE_1 and OE_2) and doxycycline treated worms (D_OE_1 and D_OE_2), and from the re-embedded blocks for control worms (RE_1 and RE_2) and doxycycline treated worms (D_RE_1 and D_RE_2).

3. I wonder whether the authors conducted a permeabilisation assay to determine the optimal time for capturing polyadenylated transcripts? If so, I believe this should be included as a supplementary file.

We thank the Reviewer for the suggestion. We did a permeabilization assay and found 2 minutes to be the optimal time. We have added **Supplementary Figure 3** (also below) to show some

example images from this permeabilization time, and added the permeabilization experiment to the Methods section under “**Tissue optimization experiment**” in lines 667-748.

Supplementary Figure 3. Tissue optimization experiment. Example images of the H&E image (left images) and corresponding cDNA fluorescent image (right images) from tissue sections from whole adult *B. malayi* female worms that were permeabilized for 2 minutes (A), 2 minutes and 10 seconds (B), and 1 minute and 30 seconds (C).

4. In line 160, the authors mention morphological annotations, but I failed to find such information. Could this be included in the figure? I believe this is an important piece of information.

This is important information to show. As stated in our response to the Results section comment #1, we added **Supplementary Figure 1** that shows, in panel D, parts of two representative sections from BM1 at larger magnification with different tissue structures annotated.

5. In figure 1C, can the authors clarify what the different numbers per sample refer to? Are these sections? If so, why there are some sections missing? The numbers do not seem consecutive (e.g., BM2_1 is missing).

Thank you for catching the worm numbering discrepancy. We have now updated the numbering of BM2 to start from 1 and end at 11 in all Figures. The section labels in **Figure 1C** consist of the worm sample (BM1, BM2, BM3), underscore, followed by the section number to label all

consecutive sections. Of note, for worm sample BM3, sections #5 and #6 are missing as these sections were discarded during cryosectioning due to poor section quality. We also updated the figure legend to clarify that these numbers indicate consecutive sections.

6. For figure 2B, can the authors generate a spatial feature plot for genes of interest identifying each of the tissue sections reported in 2C? I believe this will significantly strengthen the point made regarding the spatial identification of various tissues within the worm.

Thank you for this helpful suggestion. We showed several differentially expressed genes per cluster in 3D in the previous manuscript version Supplementary Figure 1 that we have now included in a new figure, i.e. **Figure 3** (also below), to highlight the spatial expression of these genes of interest. All genes of interest can be further explored in their 2D expression on our shiny app (<https://giacomellolabst.shinyapps.io/brugiast-shiny/>). We picked a few key marker genes for each cluster/tissue type to highlight and present as examples in our main text and corresponding **Figure 3** as suggested by the Reviewer.

Figure 3. Tissue region expression of cluster marker genes. (A) Dotplot depicting the expression of *B. malayi* cluster marker genes across the different clusters. Average expression values indicate the average UMI counts (normalized) per spot across all spots in each cluster. Percent expressed values indicate the percentage of spots within each cluster that express each gene. (B-F) Spatial expression of cluster marker genes in 3D through worm sample BM2 and

expression distribution across each cluster in violin plot. Magnification 20x. A: Anterior, P: Posterior, arrow indicates first to last section through the worm.

7. Some of the spots do not seem to be covered by tissue in figure 2B, which might be due to over-permeabilisation. Can the authors comment on this?

Thank you for raising this important point. When selecting spots to use in the analysis, any spot slightly touching the tissue was selected to maximize the amount of spatial data used in the analysis. Some spots may have appeared more outside the tissue when visualizing the spots; however, we have now updated the spot sizes when visualized on the tissue sections in all figures to be closer to their actual size on the Visium slides. In addition, the clear cuticle structure can be difficult to see in the tissue sections. The 2-minute permeabilization time was selected based on tissue optimization experiments to permeabilize the tissue sufficiently but minimize transcript diffusion (as shown in the added **Supplementary Figure 3** referred to in response to Results section comment #3).

8. Although exciting, it is unclear to me how the data reported in figure 2D was generated. I suggest that all the sections captured prior to generation of the 3D model should be provided as a separate supplementary file to make it easier to understand. This could perhaps be added to supplementary information 1.

Thank you for bringing to our attention the clarity needed around the generation of the 3D model. As mentioned in our response to Results section comments #1 and #4, the original H&E images are provided in the Mendeley dataset under Reserved DOI: [10.17632/8f62vydg3z.1](https://doi.org/10.17632/8f62vydg3z.1) that will become publicly available. The Mendeley dataset also contains alignment information on how each individual tissue section was aligned to the other sections and this information can be used to reproduce the 3D model with the corresponding worm data. The 3D model was generated by aligning tissue sections 1-9 to section 10 for worm sample BM2, as described in the Methods section under “**3D Figure**” with additional detailed information added on the 3D model construction in lines 955-964. All labeled worm sections are visible on the shiny app (<https://giacomellolabst.shinyapps.io/brugiast-shiny/>) under “Section Labels” and in the new **Supplementary Figure 1**. To see additional details about the 3D model generation, all the code is on github (https://github.com/giacomellolab/Brugia_malayi_study/tree/main/3D_model).

9. Given the importance of the conclusions presented in figure 2, the authors should validate some of their markers using alternative approaches. For instance, an in situ hybridisation approach could certainly help to demonstrate that the novel markers identified with the ST approach are indeed tissue-specific. This 2D approach would complement efforts to generate a 3D atlas.

Thank you for your comment on the importance of validation. We validated our differentially expressed genes of interest per cluster using a previously published proteomics study (Morris et al. 2015) that manually separated certain tissue structures by dissection and then detected proteins from each tissue structure (gut, etc.). Since we used validated markers per tissue from the mentioned proteomics study from comparable tissue structures to those uncovered with our clusters, we consider this analysis as a validation of these marker genes. We also used previously obtained data, such as the essential role of glycolytic enzymes (Voronin et al. 2016, 2019), to confirm our current observations. Moreover, our work expands the information about these genes

by adding the spatial resolution of the expression of glycolytic enzymes in selected regions of the worm. We have also updated the language of our manuscript to show that the genes coming up in our analysis are differentially expressed but may not necessarily be marker genes (unless they are confirmed by the previously mentioned (Morris et al. 2015) study). Furthermore, our 3D model also provides an indirect validation by showing the consistent gene expression patterns through all sections in a worm sample in the region of interest, reaffirming the location of these distinct structures across multiple sections.

Finally, while we agree that *in situ* hybridization is a valuable resource for spatial analysis, *in situ* hybridization-based approaches require extensive resources and time to develop specific probes for non-common organisms, such as *B. malayi*. Moreover, for *Brugia malayi*, there are no specific antibodies currently commercially available.

10. I am having some difficulties with figure 2E. Most of the pathways shown here are associated with ribosomal metabolism and ubiquitin, and seem to show little cluster specificity, especially for clusters 1 and 2. The pathways reported in figure 2E do not necessarily align with those mentioned in the text (e.g., I could not find sterol-binding and transfer activity in 2E). Similarly, I could not see the gene pathway associated with ShKT-domain containing proteins in the figure. Although I appreciate these are reported in the supplementary tables, perhaps including these in figure 2E would make it easier for a reader to interpret. I believe the data presented in supplementary information 1 should be moved to the main figure as it makes the point clearer. However, additional experiments validating some of these markers, either via FISH or IFA (if not reported elsewhere), should be included in the manuscript.

Thank you for bringing up the lack of clarity of Figure 2E. The processes highlighted in that figure are an overview of the statistically significant processes in each cluster across the whole dataset. In addition, we highlighted in the main manuscript text some specific genes that belong to certain processes (such as the mentioned *sterol-binding*, *transfer activity*, *ShKT-domain containing proteins*), that are statistically contributing to the GO term, but the GO term itself is not statistically significant for inclusion in **Figure 2E**. However, the function of these specific genes is important to mention. We see the confusion mentioned by the Reviewer and we have now adjusted how the Figures are referred to in the text by ensuring to only refer to **Figure 2E** for those specific processes, otherwise we refer to **Supplementary Table 4**.

We also agree that the information in Supplementary Figure 1 should be a main figure, and that information is now presented in the main text as a new **Figure 3**.

As mentioned in the response to this Reviewer's Results comment #9, for *Brugia malayi*, there are no specific antibodies currently commercially available. As it is a new field of study to spatially resolve these expression patterns on a wide scale and significantly more resources and time would be required to develop specific probes for FISH. We agree that these techniques are valuable resources for spatial analysis and hope further advances will be made on these fronts in the future.

In addition, as mentioned in our response this Reviewer's Results comment #9, we validated our differentially expressed genes of interest per cluster using a previously published proteomics study (Morris et al. 2015) that detected proteins from comparable dissected tissue structures (gut, etc.).

All genes validated from this study are shown in **Supplementary Table 3 B-F** and discussed in the manuscript lines 188-337, consisting specifically of genes Bm2854, Bm11071, Bma-hsp-17, Bma-cpt-1, Bm12794, Bm5705, Bm7941, Bm8720, Bm2751, Bm6643, Bm4112, and Bma-cht-1.2. We also used previously obtained data, such as the essential role of glycolytic enzymes (Melnikow et al. 2013; Voronin et al. 2016, 2019), to confirm our current observations for glycolysis genes Bma-ald-1, Bm5699, and Bma-ldh-1 discussed in Manuscript lines 339-375.

11. In figure 3, can the authors co-detect/correlate the expression of the selected glycolytic genes with the presence of Wolbachia? Would an imaging experiment help to substantiate the claims made in this figure?

We thank the Reviewer for the suggestion. We did not find glycolytic genes to be co-localized with *Wolbachia* with statistical significance. **Supplementary Table 6** contains the 65 differentially expressed genes between *Wolbachia*⁺ and *Wolbachia*⁻ spots that came up in our analysis, and the glycolytic genes are not among them. We have described our observation in the text of the manuscript with an accompanying explanation of our finding in the results section “**Co-localization of *B. malayi* genes in *Wolbachia*⁺ compared to *Wolbachia*⁻ regions of the tissue**” in lines 377-429.

Along similar lines to our response to comments #9 and #10, imaging based spatially-resolved transcriptomics approaches (such as FISH) are a valuable resource for spatial analysis, however, given the lack of readily available commercial reagents for these purposes, this would take extensive follow-up experiments. In this study, we aim to provide a technological advancement in developing a novel method to probe very small organisms in an unbiased, exploratory manner, such as parasitic worms, rather than large tissues and generate observational data within a specific region of interest in *Brugia malayi* adult female worms. Conducting conditional, functional experiments to look into specific genes and processes and their mechanistic details are to be done in follow-up studies. In addition, in our added experiments with antibiotic-treated worms, we observed a corresponding decrease in *Wolbachia* levels as is expected based on previous studies (as mentioned in the manuscript), providing further validation of our experimental approach.

12. In line 405 - 409: Is this statement based on a prediction from the data, or on previously reported work? I cannot seem to follow the logic of these statements. Can the authors clarify?

Yes, the statement “The presence of *Wolbachia* in the tissue is associated with a significant downregulation in expression of *B. malayi* genes that are predicted to encode proteins with functions that include protease and peptidase activity, endopeptidase activity, and proteasome-mediated protein catabolic process. This results in the potential reduction of proteolysis in *Wolbachia*⁺ tissue of female worms.” and accompanying results are a prediction from our data, specifically from our analysis of DEGs of *B. malayi* in *Wolbachia*⁺ vs *Wolbachia*⁻ spots (**Supplementary Table 6**).

We slightly rephrased the statement to improve the clarity by saying “The presence of *Wolbachia* in the tissue (determined by the detection of tRNA/rRNA bacterial transcripts) is associated with a significant downregulation in the expression of *B. malayi* genes that encode proteins annotated with functions that include protease and peptidase activity, endopeptidase activity, and

proteasome-mediated protein catabolic process (**Supplementary Table 6**). Suppression of protease activity can potentially reduce the proteolysis in *Wolbachia*+ tissue of female worms.” in lines 389-394.

13. Although I appreciate the observations and data interpretation in figure 4C and D are exciting, I believe it is a bit speculative and would suggest that the authors rephrase this section to state that this is something that needs to be experimentally validated and remains hypothetical.

Thank you for this important comment. We rephrased the manuscript text that discusses the results from the previous Figures 4C and 4D, **now Figure 5C and 5D**, to emphasize that these are hypotheses that remain to be experimentally validated, found in lines 393-394, 411,415, 423, 427-429.

14. I think it would be critical to specifically highlight which novel genes have been discovered with this approach and take them forward for validation. I appreciate this atlas will be a great resource to the community, but it is imperative to provide some kind of additional validation to support the claims, especially around genes/pathways that might have downstream implications for parasite control (e.g., interaction with Wolbachia).

Thank you for the comment. For the genes mentioned in the section “**Spatially distinct gene expression separation of digestive tract, body wall, and multiple reproductive tissue regions**”, those genes have previously been validated in the mentioned proteomics study (Morris et al. 2015) in response to Reviewer #1 Major Comment #1 and this Reviewer’s Results comments #9 and #10. The novelty presented here is the spatial tissue level analysis we can perform where we spatially resolve different structures within this region in a data-driven manner, as further stated in response to Reviewer #1 Major Comment #1. We combine the use of a commercially available platform on small filarial worms (overcoming the technical challenge due to very different length-width ratio) with untargeted, whole-transcriptome capture in intact, longitudinal tissue sections with 55 µm resolution. This approach allows us to spatially distinguish different structures, localize the *Wolbachia* endosymbiont, and compare *B. malayi* gene expression in areas with and without *Wolbachia*. Some of the genes/pathways we found in our analysis have been previously identified in the literature, as stated in response to Reviewer #1 Major Comment #1 and this Reviewer’s Results comments #9 and #10, which provides a validation of our results. Furthermore, our method facilitates a 3D reconstruction of the worm region of interest such that transcripts can be explored along multiple axes (anterior-posterior, ventral-dorsal). This 3D model also provides an indirect validation by showing the gene expression patterns throughout all sections in a worm sample region of interest to confirm the location of these distinct structures across multiple sections. We agree that certain genes/pathways that came up in our analysis will need to be more extensively studied to understand if they can work as therapeutic targets. However, such work could entail many extensive follow-up experiments across multiple studies. We think it is important that our work in this manuscript is made accessible to the research community to enable such follow-up work and new studies using the miniature-ST method.

15. In addition to the miniature ST approach, the 3D model is quite interesting, but I feel like it has not been put forward as much as it should have in the main text and figures. I would suggest

that the authors consider this point and edit the figures to incorporate some of the 3D data, as in my opinion this is one of the strongest points of the manuscript.

Thank you for your comment on emphasizing the 3D figure, and we agree that this should be put forward more in the manuscript. We have moved Supplementary Figure 1 that shows cluster marker genes in 3D to be main **Figure 3** and also emphasized the 3D model in the text.

Minor points:

- Some references are missing to support the statements (e.g., line 281-282, and again in 285 – 286, just to cite a few). Please check throughout.

Thank you for pointing this out. We have added references where previously missing in the manuscript.

Discussion

Minor points:

*- Can the authors comment on how their work compares to, or enrich previously published work reporting the use of spatial approaches to characterise antiparasitic targets in *B. malayi*?*

This is an important point. We have added two specific points to our Discussion section: (1) our work looks at a posterior region of the worm important in the *Wolbachia* - *Brugia malayi* symbiosis while a previous study using spatial approaches (Airs et al. 2022) looked at transcript distribution in serial sections of the anterior-posterior (A/P) axis of the head region of *Brugia malayi* parasites, discussed in lines 479-486 and 503-509; (2) we resolve transcripts within intact, longitudinal tissue sections at 55 μ m resolution and construct a 3D model of gene expression, as compared to the Air et al. study where they homogenize the different tissue structures within each cryosection or used laser capture microdissection (LCM) to separate the different tissue structures. By looking at another underrepresented region in parasite omics, and focusing on *Wolbachia* - *Brugia malayi* symbiosis, in addition to the benefits of separating distinct tissue structures at the gene expression level using unsupervised clustering analysis without the need for labor intensive tissue dissection by adapting a commercially available platform (10X Genomics Visium), our approach provides complementary information. We discuss the possibility of future studies readily implementing our highly usable method in the manuscript Discussion in lines 560-605.

Reviewer #3 (Remarks to the Author):

The study presented by Sounart et al is intriguing and potentially quite novel and impactful. However, I think it requires some significant clarification of various aspects and major revisions. I have structured my review into the following categories.

Thank you for your comments, we are pleased to hear the Reviewer finds the work intriguing and sees the potential novelty and impact. We have addressed the Reviewer's comments to clarify certain aspects in response to the comments below.

1. Significance and noteworthy results:

This study proposes technical and biological advances. On the technical side, the authors present a miniaturised-spatial transcriptomics approach for use with small tissue samples. However, I don't think this advance's impact is sufficiently explained. The authors mention that their samples are relatively small, but they work with 150-200uM tissue samples from worm sections that are several millimetres long. Numerous studies have applied spatial transcriptomics to study similarly sized or smaller tissues, including mouse or Drosophila embryos or C. elegans tissues.

See, for example:

*Rödelsperger, Christian, et al. "Spatial transcriptomics of nematodes identifies sperm cells as a source of genomic novelty and rapid evolution." *Molecular biology and evolution* 38.1 (2021): 229-243.*

*Peng, Guangdun, et al. "Spatial transcriptome for the molecular annotation of lineage fates and cell identity in mid-gastrula mouse embryo." *Developmental cell* 36.6 (2016): 681-697.*

*Srivatsan, Sanjay R., et al. "Embryo-scale, single-cell spatial transcriptomics." *Science* 373.6550 (2021): 111-117.*

Thank you for pointing out these exciting studies for spatially resolving different small organisms. Spatial Transcriptomics (ST) is rapidly growing and can be used along with other techniques, such as single-cell approaches.

In the first two references mentioned by the Reviewer (Rödelsperger, Christian, et al., *Molecular biology and evolution*, 2021 and Peng, Guangdun, et al., *Developmental cell*, 2016), the authors used methods, such as RNA tomography, on nematodes and mouse embryos that include a section-level mini bulk transcriptomic analysis. The transverse sections taken from the organisms along the anterior-posterior (A/P) axis were individually used for RNA-seq to provide transcriptomic data. Thus, every section was sequenced as bulk RNA-seq and as such the method does not spatially resolve the different structures and transcripts within individual sections. We were encouraged by these studies to take the next step of spatial resolution of transcriptomic data in small worms by performing spatial transcriptomics of each individual, longitudinal section (2D) and combining the datasets to construct 3D transcriptomics of selected regions of filarial worms.

The last reference suggested by the Reviewer (Srivatsan, Sanjay R., et al., 2021) worked on a mouse model by combining single-cell RNA-seq with larger spatial spots. The specific signature of the transcriptome from a single cell (from scRNA-seq) can be used to evaluate and complement ST data, however, in the case of the method in this reference, sci-Space, the spatial resolution is lower (spots of $73.2 \mu\text{m} \pm 14.1 \mu\text{m}$, about $222 \mu\text{m} \pm 7.5 \mu\text{m}$ spot-to-spot center distance) than that of our approach that uses Visium ($55 \mu\text{m}$ spot size, about $100 \mu\text{m}$ spot-to-spot center distance), making sci-Space less useful for *B. malayi* ($< 200 \mu\text{m}$ in width in longitudinal sections) or other worms that are long, but narrow in width - as compared to the mouse embryos (greater than $500 \mu\text{m}$ in width and length) used in the mentioned study. The difference in the dimensions of length compared to the width of *B. malayi* worms ($< 150 \mu\text{m}$ wide and millimeters in length) pose a unique technical challenge for spatial methods. Furthermore, single-cell isolation from *B. malayi* is simply impossible for all cell types. For example, hypodermal cells are syncytial cells and as large as a worm's length, thus our miniature-ST approach can work on intact tissue sections containing these hypodermal cells so that we can examine the transcripts in these giant cells on each section, emphasizing the advantage of our approach.

To address the Reviewer's comment, we have also cited the suggested papers and discussed them in the new version of the manuscript in the Introduction in lines 84-102 and the Discussion in lines 479-486 and 576-582.

The authors also note that head tissue spatial transcriptome has already been published for Brugia malayi. However, they do not explain why these method could not be applied to their study.

Thank you for suggesting a better discussion of the novelty of our approach. In both the Introduction in lines 72-74 and 84-93 and the Discussion in lines 479-486, we have now added discussion on how our method compares to and differs from the previous head tissue spatial transcriptome approach (Airs et al. 2022). In brief, the previous approach studies transcript distributions across the depth of serial sections collected along the anterior-posterior (A/P) axis of the head region of the worm. Although transcripts are resolved along the axis, the transcripts are not resolved within the different structures present within each individual serial section. We take a different approach of resolving transcripts within intact, longitudinal tissue sections at $55 \mu\text{m}$ resolution and separate distinct structures within each section. Sections are further combined in a 3D reconstruction of the isolated region to explore expression along all axes, anterior-posterior, ventral-dorsal (within the $55 \mu\text{m}$ resolution). In addition, we explore a different region of adult female *B. malayi* worms, a posterior region of the worm important in the *Wolbachia* - *Brugia malayi* symbiosis, and even localize *Wolbachia* within the region to explore *B. malayi* expression shifts in areas with and without *Wolbachia*. *Wolbachia* was not detected/resolved in the previous spatial transcriptome approach (Airs et al. 2022).

Biologically, Brugia malayi is a major global parasite and understanding its interactions with Wolbachia is of significant interest. However, I think the authors need to specify the significance of their study more clearly. The authors allude to this significance but they do not clearly explain it, in my opinion. For example, lines 98-100 state that 'Resolution of Wolbachia and B. malayi gene expression at the level of individual cells or tissue morphological regions is needed to design more effective therapeutic strategies for lymphatic filariasis. This may be the case, but it is not self-evident how and needs to be expanded upon. Likewise, it is not clear why and in what ways the authors' spatial transcriptomic data provide an advance over existing tissue-specific or dual

transcriptomics studies exploring Brugia and Wolbachia. Again, this isn't to say that the study isn't providing valuable new data, but rather, the need to clarify how the study provides a major biological advance, separate from the method.

We thank the Reviewer for appreciating the value of our study and for suggesting more clarity to convey its importance. We now rephrased the sentence in lines 98-100 and emphasized the advantages of our method for the current and future understanding of the biology of worms in the Introduction section of the manuscript in lines 52-66. We specifically discuss that we provide an unbiased, exploratory way to assay the whole transcriptome with spatial resolution, an advance over previous methods that used targeted approaches or bulk-level dual RNA-seq on *Brugia* and *Wolbachia*.

In the context of this study, we provide a previously unseen resolution of 55 μm across serial, longitudinal sections reconstructed in 3D to explore expression in all axes (anterior-posterior, ventral-dorsal) of the given tissue region. The biological advance is providing a reference for transcriptome-wide transcript localization patterns across the studied region and within distinct tissue structures at 55 μm resolution, a resource not previously available at this scale. Some biological insights already gleaned from the data include the higher expression of glycolytic enzyme genes in body wall regions, where higher numbers of *Wolbachia* are typically found, and that fatty acid metabolic processes are also upregulated in this region, which led us to hypothesize that *Wolbachia* could be inducing the production of glycolytic enzymes to produce the pyruvate the symbiont is dependent on *B. malayi* for and that fatty acid metabolism may be involved in providing carbohydrates to *Wolbachia* (discussed in lines 339-375 and 520-527). Additionally, our *Wolbachia* - *B. malayi* co-localization analysis showed downregulation of an SNX gene likely involved in apoptosis and autophagy-mediated elimination of apoptotic cells (Cheng et al. 2017; Norris et al. 2017) in *Wolbachia*+ tissue areas. This leads us to hypothesize that *Wolbachia* may be suppressing the late steps of autophagy and endosomal/phagosomal degradation by blocking phagosomal fusion with lysosomes to protect bacterial proteins from degradation, thus maintaining the bacterial interaction with the *B. malayi* host (discussed in lines 377-429 and 532-547). Our results already provide localized gene expression patterns not previously available and offer a resource for others to explore further.

2. Originality: To my knowledge, this is the first study to apply spatial transcriptomics to explore Brugia tissues (beyond the head tissue study the authors cited) or the interactions between Wolbachia and its host. I am unaware of anyone using the specific method the authors describe for their spatial transcriptomics experiments. However, as above, I think the authors need to explain more thoroughly what their method makes possible that existing spatial transcriptomics methods do not. In addition, I agree that the spatial data are novel, but how novel? What is now understood about this interaction that could not have been gleaned from existing studies and what is the impact of this new understanding?

As stated in response to the previous comment #1, we emphasize the advantages of our method over existing approaches with our spatial resolution (55 μm), across intact, longitudinal construction for multi-axis exploration across a 3D reconstruction of a worm region in lines 84-102, 479-486, and 576-582 of the Introduction and Discussion. In addition, this method shows how transcript expression differs within distinct tissue structures (body wall, digestive tract,

different reproductive tissue regions), where *Wolbachia* is located, and how global/transcriptome-wide *B. malayi* expression differs in areas with and without *Wolbachia*, which is not possible to analyze with bulk RNA-seq of sections. This is the first study to localize transcriptome-wide gene expression in distinct spatial compartments and thus provide a reference for transcriptome-wide transcript localization patterns across a studied region and within distinct tissue structures at 55 μm resolution, a resource that was not previously available at this scale.

To address the Reviewer's comment on the interaction between *B. malayi* and *Wolbachia*, we added an experiment of *B. malayi* worms treated with antibiotics and observed changes in *B. malayi* gene expression in *Wolbachia*+ versus *Wolbachia*- spots in doxycycline-treated worms, as compared to control worms. This analysis provided additional insights into the spatial localization and expression levels of protease genes and a heme-binding protein related genes in relation to *Wolbachia*, as discussed in the manuscript lines 431-469. In addition, as discussed in response to the previous comment #1, to our knowledge, this is the first study to use a spatial transcriptomic strategy to resolve both *B. malayi* and *Wolbachia* in intact tissue sections. First, in our study, we can separate the different tissue structures of *B. malayi* using a data-driven clustering approach based on differences in gene expression patterns. Second, we are able to localize *Wolbachia* in the tissues, compare *B. malayi* expression in areas with and without *Wolbachia*, and explore these gene expression changes in *B. malayi* expression in 3D along multiple axes (anterior-posterior, ventral-dorsal). This can be a platform for future studies that ask specific questions about host-*Wolbachia* interaction. However, we identified shifts in the expression of *B. malayi* genes involved in apoptosis, leading us to hypothesize that *Wolbachia* may contribute to worm apoptotic processes. We speculate that such genes could be promising therapeutic targets for killing adult parasites, which demonstrates the impact of this new understanding. These insights were directly gleaned from our spatial dataset and have not been previously understood from existing studies.

3. Are the conclusions supported: The technical conclusion centres around the assertion that the method produces high-quality spatial transcriptomic data from small samples. This statement is somewhat subjective. For example, how does one define a 'small' sample? I agree that the method appears to work well and suitably characterises the tissues and samples the authors have included in their study. However, whether the method is high impact depends on whether it's a major advance over existing methods. The samples the authors have assessed do not appear to be notably smaller than those analysed using existing methods for other organisms, and the authors don't explain why the method already used for Brugia head tissues is unsuitable. More detail on this would be very helpful.

Thank you for your comment. As stated in response to the previous comment #1 from this Reviewer and in response to Reviewer #1's Major concern #1 and Reviewer #2 Discussion Minor point, we added more information to the Introduction and Discussion sections in lines 84-102, 479-486, and 576-582 describing the advances of our approach over existing methods. In brief, *B. malayi* and other small parasitic worms are a different type of small sample compared to the mentioned small organisms such as mouse embryos and *Drosophila*. When considering the use of longitudinal sections through the organism, as was used in our study and in contrast to the transverse sections collected along the anterior-posterior axis in the studies mentioned previously by the Reviewer (Rödelsperger, Christian, et al., Molecular biology and evolution, 2021; Peng, Guangdun, et al., Developmental cell, 2016; Srivatsan, Sanjay R., et al., 2021), mouse embryos

and *Drosophila* have similar length-width sizes. Depending on the developmental stage, mouse embryos can also be fairly large with >500 μm sections in terms of both length and width (Srivatsan, Sanjay R., et al., 2021). Filarial worms, on the other hand, are often much smaller in width than in length, as in the case of *B. malayi* that is on average 150 μm in diameter and 43-55 mm in length for adult female worms. *B. malayi* also has the additional anatomical challenge of long, multinucleated hypodermal cord cells spanning its length, making single cell-based methods, such as the Srivatsan, Sanjay R., et al., 2021 study, unfeasible. In addition, several of the previous studies (Rödelsperger, Christian, et al., Molecular biology and evolution, 2021; Peng, Guangdun, et al., Developmental cell, 2016) resolved transcripts along a single axis, the anterior-posterior, using serial, transverse sections collected along this axis. We, however, resolve the tissue structures within individual tissue sections by resolving transcripts with 55 μm resolution within intact, longitudinal sections, which facilitates a 3D reconstruction of the analyzed worm area where transcripts can be explored along multiple axes (anterior-posterior, ventral-dorsal).

Capturing Wolbachia transcripts that are retained despite the polyA-based enrichment of the Brugia mRNAs is a clever approach. Coupling this with spatial transcriptomics allows the authors to correlate localised changes in the Brugia transcriptome and the presence or absence of the symbiont. I quite like this and think it is a novel and exciting aspect of the paper that will be of broad interest.

We are happy to hear the Reviewer finds the coupled *B. malayi* - *Wolbachia* localization work novel and exciting.

Characterising the Wolbachia transcriptome: the authors allude to being able to use their approach to undertake spatial resolution of Brugia and Wolbachia. On the latter, I think this is less clear and may require additional analysis. The authors captured Wolbachia mRNAs as by-catch. However, the assertion that they 'spatially resolve' the Wolbachia transcriptome is a bit ambiguous. If the authors mean that they were able to identify the location of Wolbachia and see how this correlated with differences in the host transcription, I agree. However, if they mean they could resolve the Wolbachia transcriptome at a spatial level, I think the current analyses can't conclude this. Maybe this is not what the authors meant. If it is, I think they need to provide more evidence. For example, they could compare the by-catch transcriptome with existing dual-RNAseq studies to see if the Wolbachia transcriptome is being replicated or look at transcript coverage metrics and other parameters commonly applied to quality control conventional RNAseq datasets.

Thank you for your comment and we apologize for the confusion. We mean that we could identify the location of *Wolbachia* by detecting rRNA and tRNA of *Wolbachia* and evaluate the expression of *Brugia* genes in regions where *Wolbachia* was detected vs *Wolbachia*-free ones. We updated the manuscript text in lines 381-390 to clarify that we did not spatially resolve the whole *Wolbachia* transcriptome, rather we spatially localized *Wolbachia* by its RNA.

4. Study design and methodology: As noted above, the study design and methodology appear sound and well-controlled. The analyses are appropriate and, except for my query regarding resolving the Wolbachia transcriptome (and my apologies if I have misunderstood here), support the study's conclusions. The methods sections are well described, and custom scripts, raw data and the

various outputs of the experiments are provided and sufficiently well described in the methods, supplementaries, and through the linked URLs the authors have provided.

Thank you for your response, we are happy to hear the Reviewer finds the study design and methodology sound and appropriate. As stated in response to the previous comment, we did not resolve the whole *Wolbachia* transcriptome, rather identified *Wolbachia* tRNA and rRNA genes that were non-specifically captured on the Visium slides and present in the sequencing reads to identify spots where *Wolbachia* are localized. In this way, we could identify spots containing *Wolbachia* (*Wolbachia*+ spots) and spots lacking *Wolbachia* (*Wolbachia*-). We apologize for the confusion, and provide more clarity on this point in the manuscript in lines 381-390.

*Overall, the approach is interesting, and the potential to apply spatial transcriptomics to study host-endosymbiont interactions is powerful and exciting. I think the study is well-executed and sound and may well be high-impact. However, this impact needs to be more thoroughly explained. The authors need to explain what their method does that current approaches cannot. Similarly, what are the overall implications of the new transcriptomic data for *Brugia* or *Wolbachia* that it provides, and does this provide major insight into the fundamental biology of the interaction or new avenues to control the parasite? Based on this, I'd prefer the article to be significantly revised but still considered for publication. I'd be happy to review a revised submission if that is required.*

We are happy to hear the Reviewer finds the work impactful for studying host-endosymbiont interactions. In terms of what our method does that current approaches cannot, current approaches are unable to provide a spatial resolution (55 μm) that can resolve the different structures within small organism tissues and explore how transcripts localize in those structures within intact tissue sections on a whole region/longitudinal view using a commercially available platform. In terms of insights into the fundamental biology of the interaction or new avenues to control the parasite, as stated in response to previous comments #1 and #2 from this Reviewer, by generating a previously unavailable resource of localized transcriptome-wide gene expression patterns within distinct spatial compartments at 55 μm resolution, we could identify localized *Wolbachia* - *B. malayi* interactions. For example, we postulated that SNX proteins (such as those predicted to be encoded by gene *Bma-snx-1*) may play a role in protecting *Wolbachia* in the cytoplasm of *B. malayi* host cells, indicating that targeting these proteins could be an avenue to explore for eliminating *Wolbachia* as a means to kill adult worms and control the parasites. Overall, we provide a proof-of-concept for the method and a resource for the community to explore, in addition to some biological insights of our own that can be further explored and validated in future works. We also added a new section on the impact of doxycycline treatment on *Wolbachia* levels in *B. malayi* tissues, where we observed an overall decrease in *Wolbachia* in treated *B. malayi* tissues compared to control worms, while certain *B. malayi* protease genes and a heme-binding protein related genes were upregulated or downregulated in *Wolbachia*+ spots compared to *Wolbachia*- spots in treated worms. Given that antibiotic-induced elimination of *Wolbachia* was previously shown to result in the death of adult worms, *B. malayi* genes impacted in tissue areas with *Wolbachia* under treatment could pose as promising therapeutic targets. We incorporate these ideas in lines 377-429, 431-469, and 532-553 in the manuscript.

References

- Airs, Paul M., Kathy Vaccaro, Kendra J. Gallo, Nathalie Dinguirard, Zachary W. Heimark, Nicolas J. Wheeler, Jiaye He, et al. 2022. "Spatial Transcriptomics Reveals Antiparasitic Targets Associated with Essential Behaviors in the Human Parasite *Brugia Malayi*." *PLoS Pathogens* 18 (4): e1010399.
- Cheng, Shiya, Kai Liu, Chonglin Yang, and Xiaochen Wang. 2017. "Dissecting Phagocytic Removal of Apoptotic Cells in *Caenorhabditis Elegans*." *Methods in Molecular Biology* 1519: 265–84.
- Combs, Peter A., and Michael B. Eisen. 2013. "Sequencing mRNA from Cryo-Sliced *Drosophila* Embryos to Determine Genome-Wide Spatial Patterns of Gene Expression." *PloS One* 8 (8): e71820.
- Luck, Ashley N., Xiaojing Yuan, Denis Voronin, Barton E. Slatko, Iqbal Hamza, and Jeremy M. Foster. 2016. "Heme Acquisition in the Parasitic Filarial Nematode *Brugia Malayi*." *FASEB Journal: Official Publication of the Federation of American Societies for Experimental Biology* 30 (10): 3501–14.
- Melnikow, Elena, Shulin Xu, Jing Liu, Aaron J. Bell, Elodie Ghedin, Thomas R. Unnasch, and Sara Lustigman. 2013. "A Potential Role for the Interaction of *Wolbachia* Surface Proteins with the *Brugia Malayi* Glycolytic Enzymes and Cytoskeleton in Maintenance of Endosymbiosis." *PLoS Neglected Tropical Diseases* 7 (4): e2151.
- Morris, C. Paul, Sasisekhar Bennuru, Laura E. Kropp, Jesse A. Zweben, Zhaojing Meng, Rebekah T. Taylor, King Chan, Timothy D. Veenstra, Thomas B. Nutman, and Edward Mitre. 2015. "A Proteomic Analysis of the Body Wall, Digestive Tract, and Reproductive Tract of *Brugia Malayi*." *PLoS Neglected Tropical Diseases* 9 (9): e0004054.
- Norris, Anne, Prasad Tammineni, Simon Wang, Julianne Gerdes, Alexandra Murr, Kelvin Y. Kwan, Qian Cai, and Barth D. Grant. 2017. "SNX-1 and RME-8 Oppose the Assembly of HGRS-1/ESCRT-0 Degradative Microdomains on Endosomes." *Proceedings of the National Academy of Sciences of the United States of America* 114 (3): E307–16.
- Peng, Guangdun, Shengbao Suo, Jun Chen, Weiyang Chen, Chang Liu, Fang Yu, Ran Wang, et al. 2020. "Spatial Transcriptome for the Molecular Annotation of Lineage Fates and Cell Identity in Mid-Gastrula Mouse Embryo." *Developmental Cell* 55 (6): 802–4.
- Srivatsan, Sanjay R., Mary C. Regier, Eliza Barkan, Jennifer M. Franks, Jonathan S. Packer, Parker Grosjean, Madeleine Duran, et al. 2021. "Embryo-Scale, Single-Cell Spatial Transcriptomics." *Science* 373 (6550): 111–17.
- Voronin, Denis, Saheed Bachu, Michael Shlossman, Thomas R. Unnasch, Elodie Ghedin, and Sara Lustigman. 2016. "Glucose and Glycogen Metabolism in *Brugia Malayi* Is Associated with *Wolbachia* Symbiont Fitness." *PloS One* 11 (4): e0153812.
- Voronin, Denis, Emily Schnall, Alexandra Grote, Shabnam Jawahar, Waleed Ali, Thomas R. Unnasch, Elodie Ghedin, and Sara Lustigman. 2019. "Pyruvate Produced by *Brugia* Spp. via Glycolysis Is Essential for Maintaining the Mutualistic Association between the Parasite and Its Endosymbiont, *Wolbachia*." *PLoS Pathogens* 15 (9): e1008085.

REVIEWERS' COMMENTS

Reviewer #1 (Remarks to the Author):

This reviewer is satisfied with the response of the authors

Reviewer #2 (Remarks to the Author):

I commend the authors for taking the time to address my concerns. I believe their manuscript is much improved and think this will be a great resource to the field of parasitic nematodes.

Reviewer #3 (Remarks to the Author):

I thank the authors for their considered responses and rebuttal of my various comments on their manuscript. These have clarified and addressed my concerns. I am happy to recommend publication and congratulate the authors on their study.

I do not need to see any further need to see a revised manuscript, however, I have two additional comments the authors may wish to address at their discretion.

1. Lines 397-406: The authors speculate on the possible reasons for the non-detection of differential transcription among the *Brugia* glycolytic genes with the presence or absence of *Wolbachia*. They propose that this could be due to variation in *Wolbachia* density across the samples. This is reasonable but, at the moment, only speculation. The relative read counts (normalised by total reads for each replicate) for each *Wolbachia*+ cluster should estimate the relative abundance of *Wolbachia* as a proxy for density, assuming each cluster is a similar physical size (which could also be adjusted for if not). The authors might consider using this to see if (a) *Wolbachia* density does indeed vary among their samples / *Wolbachia*+ subsections and, if so, (b) *Brugia* glycolytic gene transcript abundance, or any other functional group, correlates with *Wolbachia* read abundance. There is a variety of correlative gene network analysis software that could be used for this. A recent example (albeit in a very different application) can be found in Hu et al., *Sci. Adv.* 9, eadg1137 (2023) (Figure 1B), with the x-axis in the present case being inferred *Wb* density.

2. lines 431-469: The authors used doxycycline to reduce *Wolbachia* abundance in the host, allowing them to assess transcriptional changes in the worm that may be attributed to the symbiont. Using doxycycline to reduce *Wolbachia* abundance is a common approach. I think this is a nice addition, and the authors note an impact on, for example, heme-binding host genes. However, antibiotics such as doxycycline or tetracycline also impact the host cell mitochondria, disrupting mitochondrial protein synthesis and having a knock-on effect on mitochondrial metabolism and cell death markers. Researchers wishing to avoid this have looked at long-term rearing of lab-cultured hosts (not really possible for filarials), which can lead to endosymbiont loss without antibiotic intervention. This isn't possible here, but perhaps the authors might comment on how long the worms were left after doxycycline treatment to recover before the ST RNA-seq was performed. In addition, noting the comment above, it would be very powerful if the authors could show that the genes they see disrupted by doxycycline treatment also correlate with *Wolbachia* density in their non-treated samples. I don't wish to add an obstacle in the authors' path where they have added additional value to their study by including an experiment they were not required to do only to have this open up a whole new can of worms (forgive the pun!). I don't think this should prevent the manuscript from progressing with publication. Still, it may be something that the authors could look to address or at least acknowledge as a limitation that needs further exploration.

POINT-BY-POINT REPLIES TO REVIEWERS' COMMENTS

Reviewer #1 (Remarks to the Author):

This reviewer is satisfied with the response of the authors

We are happy to hear the Reviewer is satisfied with our response.

Reviewer #2 (Remarks to the Author):

I commend the authors for taking the time to address my concerns. I believe their manuscript is much improved and think this will be a great resource to the field of parasitic nematodes.

We are happy to hear the Reviewer finds the manuscript improved and also thinks the manuscript will be a valuable resource to the community. We thank the Reviewer for their comments, which we believe improved the manuscript.

Reviewer #3 (Remarks to the Author):

I thank the authors for their considered responses and rebuttal of my various comments on their manuscript. These have clarified and addressed my concerns. I am happy to recommend publication and congratulate the authors on their study.

I do not need to see any further need to see a revised manuscript, however, I have two additional comments the authors may wish to address at their discretion.

We thank the Reviewer for their comments, which we address below, and are elated to hear the Reviewer recommends our manuscript for publication.

1. Lines 397-406: The authors speculate on the possible reasons for the non-detection of differential transcription among the Brugia glycolytic genes with the presence or absence of Wolbachia. They propose that this could be due to variation in Wolbachia density across the samples. This is reasonable but, at the moment, only speculation. The relative read counts (normalised by total reads for each replicate) for each Wolbachia+ cluster should estimate the relative abundance of Wolbachia as a proxy for density, assuming each cluster is a similar physical size (which could also be adjusted for if not). The authors might consider using this to see if (a) Wolbachia density does indeed vary among their samples / Wolbachia+ subsections and, if so, (b) Brugia glycolytic gene transcript abundance, or any other functional group, correlates with Wolbachia read abundance. There is a variety of correlative gene network analysis software that could be used for this. A recent example (albeit in a very different

application) can be found in Hu et al., *Sci. Adv.* 9, eadg1137 (2023) (Figure 1B), with the x-axis in the present case being inferred *Wb* density.

We thank the Reviewer for the comment and agree that their suggested approach can give more insight into our speculation on the relationship of glycolytic genes, and other functional groups, with *Wolbachia* abundance. To address the Reviewer's point (a), we looked at the *Wolbachia* UMI counts as a proxy for *Wolbachia* density (lower UMI counts per spot indicate a lower density of *Wolbachia* and higher UMI counts per spot indicate higher density of *Wolbachia*) and how they varied across the samples and sections in our control worm dataset (**Supplementary Figure 2A-B**). Given the overall low counts of *Wolbachia*, we used raw UMI counts, as was done in other Spatial Transcriptomic papers working with spatial microbial detection (Sounart et al. 2022; Saarenpää et al. 2022). We observed that the *Wolbachia* density did vary across samples and sections (**Supplementary Figure 2A-B**). In addition, *Wolbachia* abundance varied across the *B. malayi* clusters, with cluster 3 (annotated as body wall) showing higher *Wolbachia* abundance (**Supplementary Figure 2C**). Differential expression analysis revealed the same cluster also showed higher expression of several glycolytic genes (**Figure 4A-C**), providing an initial indication of the spatial localization of *Wolbachia* and expression of certain glycolytic related genes.

Given the variation in *Wolbachia* abundance, we next addressed the Reviewer's point (b) by utilizing the approach mentioned by the Reviewer (Hu et al. 2023)), that was also used in several recent studies working with transcriptomic (Almudi et al. 2020; Martín-Zamora et al. 2023) and spatial transcriptomic data (Mohenska et al. 2022; Xia et al. 2022). We binned the *Wolbachia* UMI counts per spot into four groups (None, Low, Medium, and High) and calculated the average expression for each *B. malayi* gene across all the spots within each *Wolbachia* abundance group. We then used the MFuzz (Kumar and E Futschik 2007) soft clustering approach to explore *B. malayi* gene expression patterns across different densities of *Wolbachia*. We identified four soft clusters (termed Patterns to avoid confusion with our original clusters) that contain core cluster genes that peak at different *Wolbachia* abundance groups (**Supplementary Figure 2D**). In Pattern 3, the core genes that peak, or show higher expression changes, at the highest *Wolbachia* abundance group contain three glycolysis related genes (Bma-enol-1, Bma-hxk-1.1, Bm9363) (**Supplementary Figure 2D, Supplementary Data 6C**). Therefore, we see an indication that *B. malayi* glycolytic gene transcript abundance from several glycolysis associated genes correlates with *Wolbachia* abundance. In addition, we used topGO (Alexa, Rahnenführer, and Lengauer 2006) to perform Gene Ontology (GO) term enrichment analysis of each Pattern and identify the top biological processes enriched within each Pattern (**Supplementary Figure 2E, Supplementary Data 7A-D**). Pattern 3 shows an enrichment of GO terms related to stress reactions, including ROS, oxidative stress, and response to cellular toxins (**Supplementary Figure 2E, Supplementary Data 7C**). We hypothesize that the bacteria may be responsible for initiating these reactions in the host cells. Future work could aim to

directly target all *Wolbachia* genes, which could improve the detection of *Wolbachia* and give a more nuanced picture of *Wolbachia* density in different areas of the *B. malayi* tissue.

2. lines 431-469: The authors used doxycycline to reduce Wolbachia abundance in the host, allowing them to assess transcriptional changes in the worm that may be attributed to the symbiont. Using doxycycline to reduce Wolbachia abundance is a common approach. I think this is a nice addition, and the authors note an impact on, for example, heme-binding host genes. However, antibiotics such as doxycycline or tetracycline also impact the host cell mitochondria, disrupting mitochondrial protein synthesis and having a knock-on effect on mitochondrial metabolism and cell death markers. Researchers wishing to avoid this have looked at long-term rearing of lab-cultured hosts (not really possible for filarials), which can lead to endosymbiont loss without antibiotic intervention. This isn't possible here, but perhaps the authors might comment on how long the worms were left after doxycycline treatment to recover before the ST RNA-seq was performed. In addition, noting the comment above, it would be very powerful if the authors could show that the genes they see disrupted by doxycycline treatment also correlate with Wolbachia density in their non-treated samples. I don't wish to add an obstacle in the authors' path where they have added additional value to their study by including an experiment they were not required to do only to have this open up a whole new can of worms (forgive the pun!). I don't think this should prevent the manuscript from progressing with publication. Still, it may be something that the authors could look to address or at least acknowledge as a limitation that needs further exploration.

To the Reviewer's point on how long the worms were left after doxycycline treatment, worms were treated with doxycycline (12.5 μ M) for 3 days *in vitro*, and after treatment, the worms were immediately washed with PBS (3 times) and embedded within OCT. Then, the blocks were frozen and kept at -80°C until tissue sectioning, as is standard for performing ST. We added this information to the Methods.

To address the second point made by the Reviewer to explore if genes disrupted by doxycycline treatment correlate with *Wolbachia* density in the non-treated (control) samples, first, these genes were found to be differentially expressed between *Wolbachia*⁺ and *Wolbachia*⁻ spots in the non-treated samples, indicating these genes show transcriptional changes that colocalize with *Wolbachia* (**Figure 5B, Supplementary Data 5**). Second, in the new soft clusters (Patterns) generated by the approach described in response to the previous comment #1, we observed several of these DE genes between *Wolbachia*⁺ and *Wolbachia*⁻ spots in the non-treated samples (Bm17156, Bm294, Bm16, Bma-bbs-8, Bma-rpl-33.1, Bm6111, Bm13981, Bm8720) were also among the genes found to correlate with *Wolbachia* density (**Supplementary Figure 2D, Supplementary Data 6A-C**). Both points strengthen our observations of the *B. malayi* transcriptional changes in relation to *Wolbachia*, which we note in the manuscript section "Spatial transcriptomic patterns in post-doxycycline treated worms". However, as the Reviewer

points out, we are unable to completely separate the effects of *Wolbachia* reduction from the direct effects of doxycycline treatment on the worm. This is a limitation of the study, and future work could aim to give the worms more recovery time after doxycycline treatment to minimize its effect unrelated to *Wolbachia* depletion. We added a brief explanation of this limitation to our Discussion.

Supplementary Figure 2. *B. malayi* gene expression correlation with *Wolbachia* abundance variation. A-C Violin plots of the *Wolbachia* unique molecules (UMIs) per spot (raw) across the

different worm samples used in the study (**A**), the different worm sample sections (sample name_consecutive section #) used in the study (**B**), and the *B. malayi* clusters (**C**). **D**. MFuzz soft clusters (Patterns 1-4) of *B. malayi* genes co-expressed across different *Wolbachia* abundance groups. **E**. Bar plots of unadjusted *p*-values (*p*) of the top enriched Gene Ontology (GO) terms for each pattern of *B. malayi* co-expressed transcripts calculated from a two-sided Fisher's exact test.

References

- Alexa, Adrian, Jörg Rahnenführer, and Thomas Lengauer. 2006. "Improved Scoring of Functional Groups from Gene Expression Data by Decorrelating GO Graph Structure." *Bioinformatics* 22 (13): 1600–1607.
- Almudi, Isabel, Joel Vizueta, Christopher D. R. Wyatt, Alex de Mendoza, Ferdinand Marlétaz, Panos N. Firbas, Roberto Feuda, et al. 2020. "Genomic Adaptations to Aquatic and Aerial Life in Mayflies and the Origin of Insect Wings." *Nature Communications* 11 (1): 2631.
- Hu, Liang, Xiaoling Liu, Qiaolan Zheng, Wuhe Chen, Hao Xu, Hengrui Li, Jiaxin Luo, et al. 2023. "Interaction Network of Extracellular Vesicles Building Universal Analysis via Eye Tears: iNEBULA." *Science Advances* 9 (11): eadg1137.
- Kumar, Lokesh, and Matthias E Futschik. 2007. "Mfuzz: A Software Package for Soft Clustering of Microarray Data." *Bioinformatics* 2 (1): 5–7.
- Martín-Zamora, Francisco M., Yan Liang, Kero Guynes, Allan M. Carrillo-Baltodano, Billie E. Davies, Rory D. Donnellan, Yongkai Tan, et al. 2023. "Annelid Functional Genomics Reveal the Origins of Bilaterian Life Cycles." *Nature* 615 (7950): 105–10.
- Mohenska, Monika, Nathalia M. Tan, Alex Tokolyi, Milena B. Furtado, Mauro W. Costa, Andrew J. Perry, Jessica Hatwell-Humble, et al. 2022. "3D-Cardiomics: A Spatial Transcriptional Atlas of the Mammalian Heart." *Journal of Molecular and Cellular Cardiology* 163 (February): 20–32.
- Saarenpää, Sami, Or Shalev, Haim Ashkenazy, Vanessa de Oliveira-Carlos, Derek Severi Lundberg, Detlef Weigel, and Stefania Giacomello. 2022. "Spatially Resolved Host-Bacteria-Fungi Interactomes via Spatial Metatranscriptomics." *bioRxiv*. <https://doi.org/10.1101/2022.07.18.496977>.
- Sounart, Hailey, Enikő Lázár, Yuvarani Masarapu, Jian Wu, Tibor Várkonyi, Tibor Glasz, András Kiss, et al. 2022. "Dual Spatially Resolved Transcriptomics for SARS-CoV-2 Host-Pathogen Colocalization Studies in Humans." *bioRxiv*. <https://doi.org/10.1101/2022.03.14.484288>.
- Xia, Keke, Hai-Xi Sun, Jie Li, Jiming Li, Yu Zhao, Lichuan Chen, Chao Qin, et al. 2022. "The Single-Cell Stereo-Seq Reveals Region-Specific Cell Subtypes and Transcriptome Profiling in Arabidopsis Leaves." *Developmental Cell* 57 (10): 1299–1310.e4.